# Population-scale gene-based analysis of whole-genome sequencing provides insights into metabolic health

Yajie Zhao [1,2,3,12], Sam Lockhart [4,5,12], Jimmy Liu [6,12], Xihao Li [7,8,12], Adrian Cortes [2], Xing Hua[9,10], Eugene J. Gardner [1], Katherine A. Kentistou [1], Marisa Cañadas-Garre[11], Laurie Fabian [11], Karen Ho[11], Nicholas Timpson [11], Yancy Lo [6], Jonathan Davitte[6], David B. Savage [4], Carolyn Buser-Doepner[6], Ken K. Ong[1], Haoyu Zhang [9,12], Robert Scott [2,12], Stephen O'Rahilly [4,12] & John R. B. Perry [1,4,12] ✉

In addition to its coverage of the noncoding genome, whole-genome sequencing (WGS) may better capture the coding genome than exome sequencing. Here we sought to exploit this and identify new rare, protein-coding variants associated with metabolic health in WGS data ($n$ = 708,956) from the UK Biobank and All of Us studies. Identified genes highlight new biological mechanisms, including protein-truncating variants (PTVs) in the DNA double-strand break repair gene *RIF1* that have a substantial effect on body mass index (2.66 kg m$^{-2}$, s.e. 0.43, $P$ = 3.7 × 10$^{-10}$). *UBR3* is an intriguing example where PTVs independently increase body mass index and type 2 diabetes risk. Furthermore, PTVs in *IRS2* have a substantial effect on type 2 diabetes (odds ratio 6.4 (3.7–11.3), $P$ = 9.9 × 10$^{-14}$, 34% case prevalence among carriers) and were also associated with chronic kidney disease independent of diabetes status, suggesting an important role for IRS2 in maintaining renal health. Our study demonstrates that large-scale WGS provides new mechanistic insights into human metabolic phenotypes through improved capture of coding sequences.

Genome-wide, hypothesis-free interrogation of the association between genomic variants and human traits and diseases in large populations has resulted in many key insights into the pathogenesis of common cardiometabolic disorders. The power of this approach has increased with the availability of population-scale whole-exome sequencing (WES) data[1]. In contrast to earlier common-variant genome-wide association studies (GWAS), where most associated variants are noncoding[2–4] and the causal gene is often unclear, studies leveraging rare protein-coding variation in gene-based collapsing tests more confidently identify causal genes and directions of effect relative to gene function. This approach more readily identifies new causal pathways and mechanisms of disease for experimental interrogation[5].

A recent advance has come from the widespread adoption of WGS in large population studies[6]. Although the obvious advantage of WGS above WES is its ability to interrogate the noncoding genome, it has also been demonstrated that WGS identifies more functional coding variation than exome sequencing based technologies[7].

Here we sought to leverage the increased sample size and purported enhanced capture of rare coding variation from UK Biobank (UKBB) WGS data[7] to provide new insight into the genetic basis of two cardiometabolic traits of principal significance to population health: type 2 diabetes (T2D) and body mass index (BMI). Previous large-scale WES studies have identified several genes harboring rare protein-coding variants of large effect for these traits[8–12], including

examples where heterozygous loss of function either increases (for example, *GIGYF1* for T2D[13], *BSN* for obesity[9,14]) or decreases the risk of disease (for example, *MAP3K15* for T2D[12], *GPR75* for obesity[8]). By extending these analyses to consider WGS data in about 490,000 UKBB participants, we identify five new associations that we replicate in around 220,000 individuals from the All of Us (AoU) study. These findings include T2D risk-increasing PTVs in *IRS2*, encoding a key node in the insulin/IGF1 signaling cascade, which also increased risk of chronic kidney disease (CKD) independent of diabetes status, and PTVs in the ubiquitin ligase gene *UBR3* with independent effects on BMI and T2D risk. Together, these findings identify new genetic determinants of cardiometabolic risk and highlight impaired IRS2-mediated signaling as an unexpected candidate mechanism of renal disease.

## Results

To identify genes associated with either adult BMI or T2D risk, we performed association testing using WGS data available in up to 489,941 UKBB participants (Methods). This represents a sample size increase of up to 71,505 people compared to our recent WES analyses of the same cohort[9,11], attributable to both an increase in the number of sequenced samples ($n = 35,725$) and the inclusion of people of non-European ancestry ($n = 64,609$). Individual gene-burden tests were performed by collapsing rare (minor allele frequency (MAF) <0.1%) variants across 19,457 protein-coding genes. We tested three categories of variants based on their predicted functional impact: high-confidence PTVs, and two overlapping missense masks that used a rare exome variant ensemble learner[15] (REVEL) score threshold of 0.5 or 0.7. This yielded a total of 81,350 tests (40,750 tests for T2D and 40,600 tests for BMI) for gene masks with at least 30 informative rare allele carriers, corresponding to a conservative multiple-test corrected statistical significance threshold of $P < 6.15 \times 10^{-7}$ (0.05 of 81,350).

Genetic association testing identified a total of 21 genes with at least one mask associated at this threshold with adult BMI ($n = 10$ genes) or T2D ($n = 12$ genes) (Fig. 1 and Supplementary Table 1). The only overlapping association between the two traits was with PTVs in *UBR3*. Our WGS analysis confirmed previously reported gene associations using WES for BMI, including PTVs and damaging missense variants in *MC4R*, *UBR2*, *SLTM* and *PCSK1*, *BSN*, *APBA1* and *PTPRG*[8,10,13,14]. Our WGS analysis also confirmed previously reported gene associations using WES for T2D including PTVs in *GCK*, *HNF1A*, *GIGYF1* and *TNRC6B*, and missense variants with REVEL >0.7 in *IGF1R*[11,13]. Our WGS gene-burden test seemed statistically well calibrated, as indicated by low exome-wide test statistic inflation ($\lambda_{GC} = 1.15$ for BMI and 1.20 for T2D) and by the absence of significant associations with any synonymous variant masks (included as a negative control).

### Identification of new genetic risk factors for BMI

At the three genes that we newly identified for BMI, PTVs conferred higher adult BMI: *RIF1*, encoding an effector in the nonhomologous end-joining pathway activated in response to double-stranded DNA-breaks[16]; *UBR3*, encoding an E3-ubiquitin ligase that is highly expressed in sensory tissues[17] and the nonreceptor tyrosine kinase gene *TNK2*. Previous GWAS also identified loci associated with BMI within 500 kb of *TNK2* (BMI: rs34801745:C, beta = 0.013, s.e. = 0.002, $P = 7 \times 10^{-11}$) (Supplementary Table 2). Using a variant to gene mapping method[18] (Methods), GWAS signals at both the *TNK2* loci could be confidently linked to the function of this gene; for example, we observed colocalization between expression quantitative trait loci for *TNK2*, with decreased expression corresponding to increased BMI, directionally concordant with their rare PTV effects (Supplementary Table 2).

### Identification of new genetic risk factors for T2D

At the seven genes that have not been implicated previously by population-scale studies for T2D, PTVs conferred higher risk for T2D: *IRS2*, encoding a key adapter molecule in the insulin-signaling cascade;

*UBR3*, encoding an E3-ubiquitin ligase that is highly expressed in sensory tissues[17]; *NAA15*, encoding a component of N-terminal acetyltransferase complexes[19] and *RMC1*, encoding part of a protein complex critical for lysosomal trafficking and autophagy[20,21] (Supplementary Table 1). Our missense mask also identified associations with *IP6K1*, encoding an inositol phosphokinase, the known MODY gene *HNF4A* and *UBB* encoding ubiquitin (Supplementary Table 1). There were also common GWAS loci associated with T2D within 500 kb of *IRS2* (T2D: rs9301365: T, beta = 0.024, s.e. = 0.003, $P = 2.1 \times 10^{-16}$), *RMC1* (T2D: rs1788819:G, beta = 0.032, s.e. = 0.003, $P = 4 \times 10^{-21}$), *IP6K1* (T2D: rs7613875:A, beta = 0.025, s.e. = 0.003, $P = 4.8 \times 10^{-16}$) and *HNF4A* (T2D: rs12625671:C, beta = 0.067, s.e. = 0.004, $P = 1.7 \times 10^{-68}$) (Supplementary Table 2). Of these, we could confidently link variants at the *IRS2* and *HNF4A* T2D loci with the corresponding gene's function (Supplementary Table 2).

### Sensitivity analyses

We also tested whether any of our rare variant discoveries were 'tagged' by common-variant associations. We generated polygenic risk scores for each trait and included these as covariates. Of our five confirmed new gene-disease associations, four were modestly attenuated (but retained exome-wide significance) and one was modestly strengthened (*HNF4A* T2D). We performed additional analyses adjusting for regional, common single-variant associations identified directly in UKBB WGS data (MAF > 0.001, $P < 6.15 \times 10^{-7}$) and did not observe any meaningful attenuation in test statistics (Supplementary Tables 20 and 21). These results indicate that the identified rare variant effects on T2D and BMI are independent of common variants (Supplementary Table 3). In addition, as we identified common genetic variation probably acting through *IRS2* to be associated with T2D, these results support the presence of a genuine allelic series supporting the role of *IRS2* in T2D risk.

As a further sensitivity analysis, we performed 'leave-one-out analyses,' which confirmed that none of the above gene-level associations were driven by a single rare variant (Supplementary Table 4). Furthermore, all new associations exhibited similar effects in published results using WES data from UKBB but at subthreshold significance ($P \leq 8.3 \times 10^{-5}$).

### Increased power using an all-ancestry WGS approach

For most of the associated genes, we observed stronger associations using WGS than we reported previously using WES, with an overall 29% increase in mean chi-square values for these associated genes using similar variant masks (Supplementary Table 5). To ascertain the determinants of this stronger association, we first compared the effect sizes in the current study and our previously published whole-exome analyses ($n_{BMI} = 419,668$ and $n_{T2D} = 418,436$), observing comparable effect sizes (Supplementary Fig. 1). Next, we examined the increase in sample size in our all-ancestries based approach to a European-only WGS analysis using otherwise identical analytical parameters. Among the 27 significant associations we identified, 21 had a stronger $P$ value in the all-ancestries sample, with a 4.6% increase in mean chi-square values. To similarly quantify the gain in statistical power using WGS, in the UKBB sample with both WGS and WES data available, WGS produced a 21% increase in mean chi-square values for the associated genes masks (Supplementary Table 5) and included (median = 11.5, quartile 1 to quartile 3: 4.5–16.5) more variants compared to WES. Moreover, sensitivity gene-burden tests considering only those additional carriers identified by WGS (that is, not identified by WES data), 16 of the 23 gene masks with at least five carriers showed a nominally significant association ($P < 0.05$) with the target phenotype, indicating that the additional coding variants identified by WGS are likely to be functionally relevant. In contrast, gene masks (with at least five carriers) of WES-only variants did not show even nominally significant associations with the target phenotype. To exemplify the benefits of WGS versus WES based sequencing in UKBB, we show *IRS2* coding variant detection by WES and WGS in Supplementary Figure 2. Restricting analysis to samples with

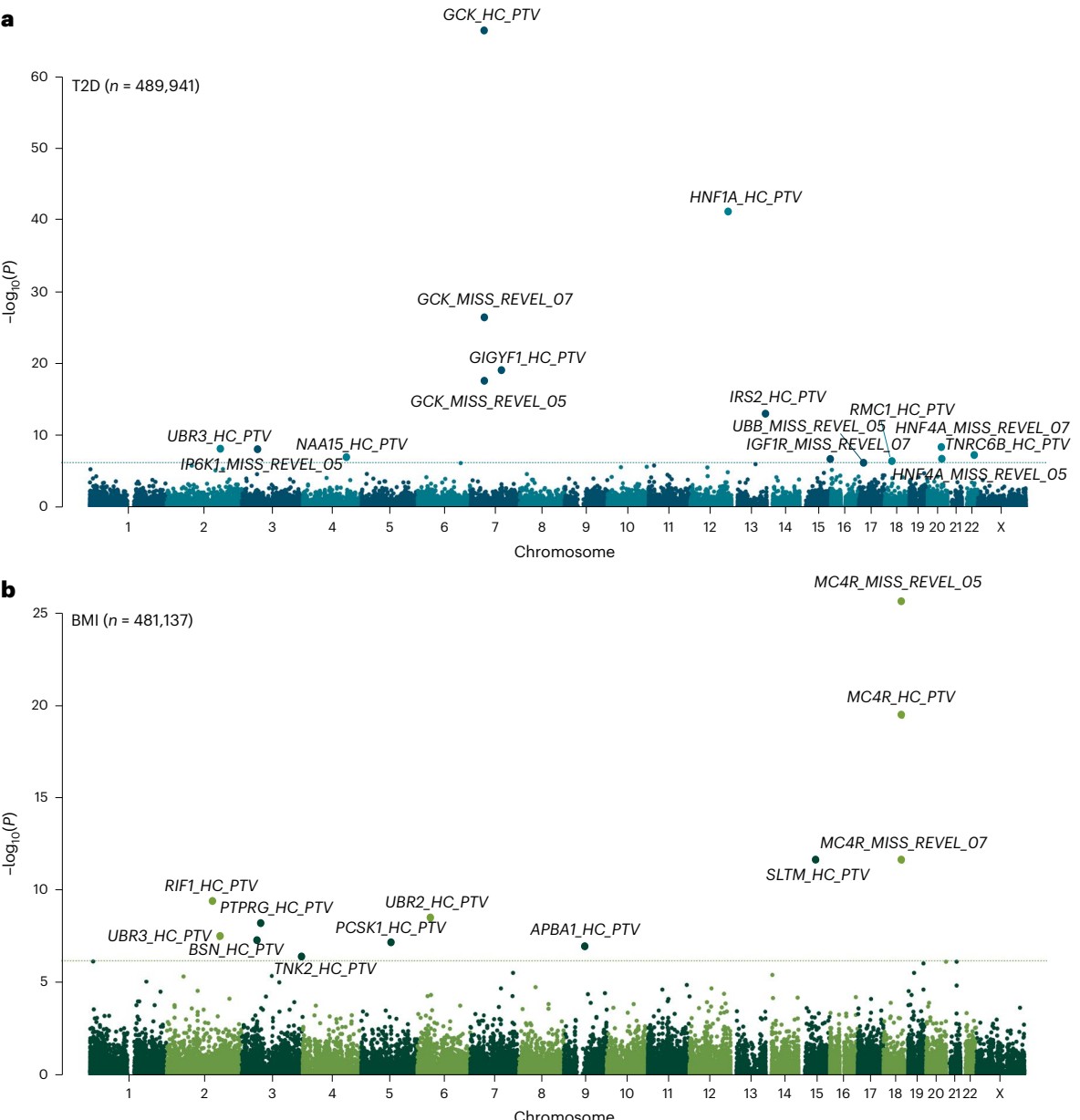

**Fig. 1 | Genome-wide multi-ancestry gene-burden test for T2D and BMI in UKBB. a,b,** Manhattan plots showing gene-burden test results for T2D (**a**) and BMI (**b**) with unadjusted two-sided *P* values derived from gene-burden testing conducted in BOLT-LMM and plotted on a −log₁₀ scale. Genes passing exome-wide significance ($P < 6.15 \times 10^{-7}$ (0.05/81,350)) are labeled. Points are annotated with variant mask information. *MISS_REVEL*, missense variants with REVEL scores above 0.5 or 0.7; *HC_PTV*, high-confidence PTVs.

both WGS and WES sequencing of *IRS2*, WGS identifies 15 more PTVs than WES (79% increase), resulting in an almost 50% increase in sample size for this mask (Supplementary Table 5). Increased variant discovery did not seem to be restricted to select regions of *IRS2* (Supplementary Fig. 2). Our findings confirm and quantify the enhanced coverage of coding variants provided by WGS above WES in UKBB.

### Replication in AoU of five new genes associated with BMI and/or T2D

To replicate our findings in UKBB WGS data, we implemented an identical variants annotation workflow for genes identified from UKBB and ran gene-burden testing using WGS data derived from 219,015 participants in the AoU studies. Two of the three new gene associations with BMI (*RIF1* and *UBR3*) were replicated in AoU (at *P* < 0.05; Fig. 2 and Supplementary Table 6), whereas three of the seven gene associations with T2D (*IRS2*, *UBR3* and *HNF4A*) were replicated in AoUs

(Supplementary Table 6). All of these associations remained significant (*P* < 0.05) after adjustment for BMI (Supplementary Table 7).

To understand whether failed replication was related to limited statistical power, we conducted power calculations after correction for winner's curse (Methods). For BMI, the risk of type 2 error exceeded 30% for all three of the nonreplicating masks. For T2D, four of the nonreplicating masks had a type 2 error rate exceeding 15%, whereas the other three nonreplicating masks had adequate power (*GCK* Missense, REVEL >0.5, *TNRC6B* and *NAA15* PTVs) (Supplementary Tables 8 and 9).

### A phenotypic association scan reveals a role for *IRS2* in human kidney health

To explore the broader phenotypic effects of our identified BMI-raising and T2D risk genes, we conducted a phenotypic association scan (PheWAS) for each gene variant mask significantly associated with T2D and BMI in our discovery analysis (Supplementary Tables 10 and 11).

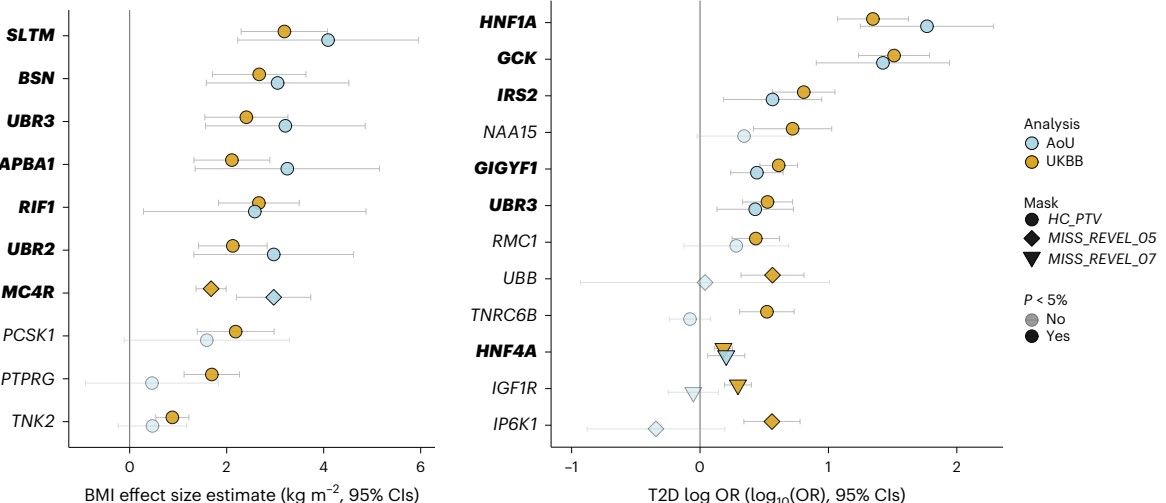

**Fig. 2 | Discovery and replication of significant associations with BMI and T2D in UKBB and AoU.** Plots show effect estimates for predicted damaging mutations in the indicated gene on BMI (left) and T2D risk (right) in the UKBB and AoU. In UKBB, effect estimates for BMI were derived using GLMs. In AoU, effect estimates were approximated from score statistics and their variances under a GLM framework. $n_{BMI, UKBB} = 481,137$; $n_{BMI, AoU} = 219,015$; $n_{T2D, UKBB} = 489,941$; $n_{T2D, AoU} = 219,015$. Odds of T2D are plotted on a $\log_{10}$ scale. All error bars represent 95% CIs and all P values are two-sided. Gene names of results replicated in AoU are highlighted in bold.

We observed several expected associations, for example, between T2D risk genes with HbA1c and glucose and between BMI genes with whole body fat mass (Fig. 3). However, we were intrigued to observe a new, highly statistically significant association of *IRS2* PTVs with lower Cystatin-C-derived estimated glomerular filtration rate (eGFR; effect = −12.92 ml⁻¹ min 1.73 m⁻², s.e. = 1.87, P = 4.9 × 10⁻¹², carrier n = 55). This effect of *IRS2* PTVs on renal function was consistently observed across three different methods of GFR estimation (Fig. 4). This association does not simply reflect the consequences of T2D-mediated chronic hyperglycemia on renal function as it was also observed in carriers of PTVs in *IRS2* without a diagnosis of T2D (Cystatin-C-derived eGFR: effect = −10.42 ml⁻¹ min 1.73 m⁻², s.e. = 2.24, P = 3.3 × 10⁻⁶, carrier n = 36), and effects were still observed after including T2D status as a covariate in the model (Supplementary Table 12). Consistent with a renoprotective role for IRS2 in humans, PTVs in *IRS2* were associated with an increase of around fourfold in odds of CKD (odds ratio (OR) = 4.0, 95% confidence interval (CI) (1.9–8.6), P = 3.1 × 10⁻⁴, carrier n = 58, 14% case prevalence; Fig. 4). Again, this association persisted after adjustment for diabetes status (Supplementary Table 12). Finally, we sought to demonstrate the robustness of this observation with orthogonal validation in an independent cohort. Therefore, we undertook a lookup of *IRS2* in a publicly accessible PheWAS of the AoU Cohort (Methods) and found nominally significant, highly ranked associations for a biomarker of renal function (blood urea nitrogen), CKD and other traits related to renal failure (Supplementary Table 13). These results identify *IRS2* as a T2D risk gene with an independent effect on CKD risk.

We also observed that PTVs in the adapter protein *GIGYF1* conferred beneficial effects on serum lipids, consistent with previous findings[22], but deleterious effects on renal function, including a roughly twofold increase in odds of CKD (Fig. 3 and Supplementary Table 10). We also note a striking reduction in circulating SHBG (sex hormone binding globulin) levels in carriers of predicted damaging missense mutations in *HNF4A* (effect = −6.4 nmol l⁻¹, s.e. = 0.73, P = 7.5 × 10⁻¹⁹, carrier n = 1,200), which has been reported to regulate *SHBG* transcription in vitro[23]. PTVs in *RMC1* were associated with higher triglycerides, lower high-density lipoprotein (HDL) (and therefore higher triglyceride (TG):HDL ratio) and increased risk of metabolic (dysfunction)-associated fatty liver disease—a pattern suggestive of lipotoxic insulin resistance.

## Evidence of functional diversity in IRS1/IRS2-mediated signaling

IRS1 and IRS2 are critical nodes in the insulin/IGF1 signaling cascade. They are recruited to, and phosphorylated by, the activated insulin receptor, serving as essential adapter molecules to mediate downstream signaling. An interesting finding from mouse genetic studies is that *Irs1* knockout mice do not show fasting hyperglycemia, despite evidence of insulin resistance and reduced body size, consistent with impaired growth due to IGF1[24,25]. In contrast, *Irs2* knockout mice are comparable in size to their littermate controls but exhibit fasting hyperglycemia and glucose intolerance due to failed beta-cell compensation[26]. To determine whether similar phenotypic heterogeneity is present in humans, we compared the effects of *IRS1* and *IRS2* loss of function mutations (Fig. 5 and Supplementary Tables 10 and 11). Consistent with the described mouse biology, human carriers of PTVs in *IRS1* had reduced fat-free mass and reduced height, suggestive of impairment in the anabolic effects of IGF1 signaling. In contrast, carriers of PTVs in *IRS2* had no changes in lean mass or height, but a substantially increased risk of T2D (Fig. 5). These findings suggest that the functional specificity of IRS1/IRS2 described previously in mice is conserved in humans; IRS1 probably mediates the effects of IGF1 signaling on linear growth and lean mass, whereas IRS2 is relatively more important for glucose tolerance, probably due to its key regulatory actions in the pancreatic beta cell.

## No evidence for a highly penetrant severe insulin resistance syndrome in carriers of *IRS2* PTVs

Damaging mutations in canonical members of the insulin signaling cascade can cause highly penetrant monogenic severe insulin resistance syndrome, but there is considerable phenotypic heterogeneity. For example, dominant-negative mutations in *INSR* cause a monogenic severe insulin resistance syndrome that often presents in adolescence/early adulthood, whereas simple loss-of-function mutations carried in heterozygosity do not cause severe insulin resistance, but probably increase risk of T2D in later life[27,28]. It is plausible that the T2D risk associated with *IRS2* PTVs is a manifestation of a severe insulin resistance syndrome in these carriers. Unfortunately, we cannot test this in UKBB as insulin measurements are unavailable, and surrogate measurements of insulin resistance are not reliable when the affected gene is proximal in the insulin signaling pathway. Therefore, to assess whether

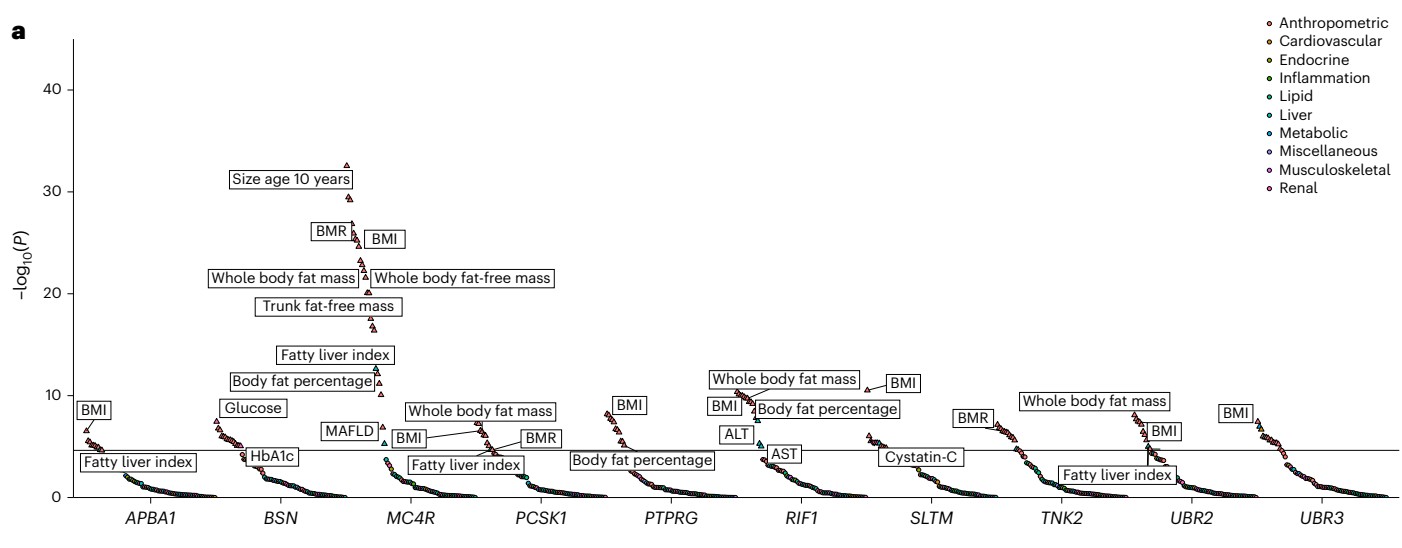

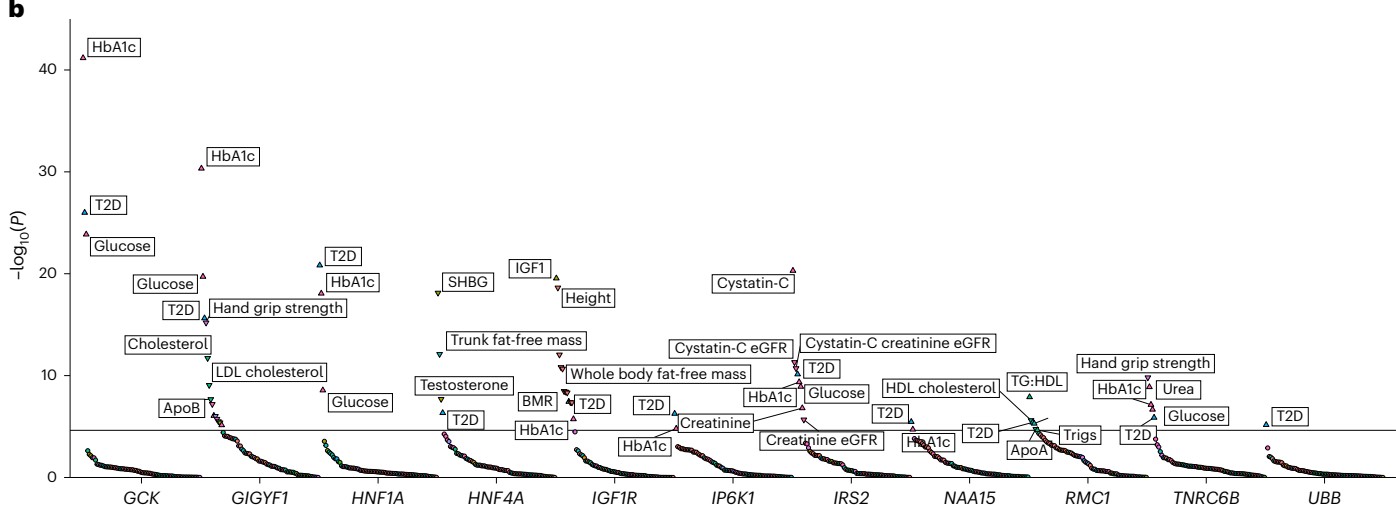

**Fig. 3 | PheWAS of BMI and T2D associated genes in UKBB. a,b,** Effects of the most significant Gene × Mask association with BMI (**a**) or T2D (**b**) were assessed (Methods) on a panel of 79 traits, and resulting *P* values were plotted on a −log₁₀ scale. *P* values are two-sided and unadjusted. Test statistics were derived from linear and logistic regression models performed using the GLM framework. Numbers of participants are provided in Supplementary Tables 10 and 11. Points are colored according to classification of phenotype; the orientations of triangles indicate the direction of effect for significant traits. For clarity, only a subset of traits and the most significant Gene × Mask association (for genes with more than one mask significantly associated with T2D or BMI) are displayed. *UBR3*, which was associated with both T2D and BMI in our discovery analysis, is presented alongside BMI risk genes only to avoid duplication. The solid horizontal lines represent a Bonferroni-corrected threshold for statistical significance of $2.35 \times 10^{-5}$ (0.05/2,132 Phenotype × Mask associations).

*IRS2* PTVs may cause a severe monogenic insulin resistance syndrome, we interrogated exome sequence data from The Avon Longitudinal Study of Parents and Children (ALSPAC)[29]—a birth cohort with fasting insulin measurements available in a substantial subset. We found two carriers of *IRS2* PTVs, both of whom had normal (between 2.5th and 97.5th of age and sex matched centiles) serum insulin levels in late adolescence/early adulthood (Supplementary Tables 14 and 15). Thus, it seems unlikely that *IRS2* PTVs cause a penetrant monogenic severe insulin resistance syndrome.

### E3-ubiquitin ligases UBR2 and UBR3, body composition and cardiometabolic risk

UBR2 and UBR3 are related E3-ubiquitin ligases. UBR2 is a canonical N-recognin that recognizes modified N-terminal amino acid residues (so-called N-degrons) and ubiquitinates these proteins to target them for degradation[30,31]. UBR3 shares weak homology with UBR2. Although UBR3 does not possess N-recognin activity, it does mediate N-terminal ubiquitinition through an as yet unknown degradation signal[30,32]. In our discovery analyses, *UBR2* and *UBR3* PTVs were both associated with increased BMI, but only *UBR3* conferred a significant increase in T2D risk (Figs. 1 and 3), consistent with distinct molecular actions of the encoded proteins. The association of PTV in *UBR3* and T2D was not solely due to increased BMI as the effect on T2D was attenuated only partially after adjustment for BMI (OR = 2.5, 95% CI (1.5−4.1), $P = 2.7 \times 10^{-4}$). To gain further insight into the mechanism through which UBR3 disruption increases T2D risk, we examined associations with body composition and surrogate markers of insulin resistance measured in UKBB, SHBG and TG:HDL (Supplementary Table 11). We found no evidence for an effect of PTVs in *UBR3* on body fat distribution as assessed by WHRadjBMI and inconsistent effects on the surrogate markers of insulin resistance, that is, TG:HDL was not altered but SHBG was nominally decreased.

UBR2 has been implicated in regulation of muscle mass in several mouse studies[33–35]. Therefore, we assessed the effect of *UBR2* and *UBR3* PTVs on lean and fat mass measured by bioimpedance in UKBB. Carriage of a PTV in *UBR2* or *UBR3* was associated with higher whole body fat mass and, whereas *UBR2* PTV carriers showed a nominal increase in whole body fat-free mass, this association was modest and likely to

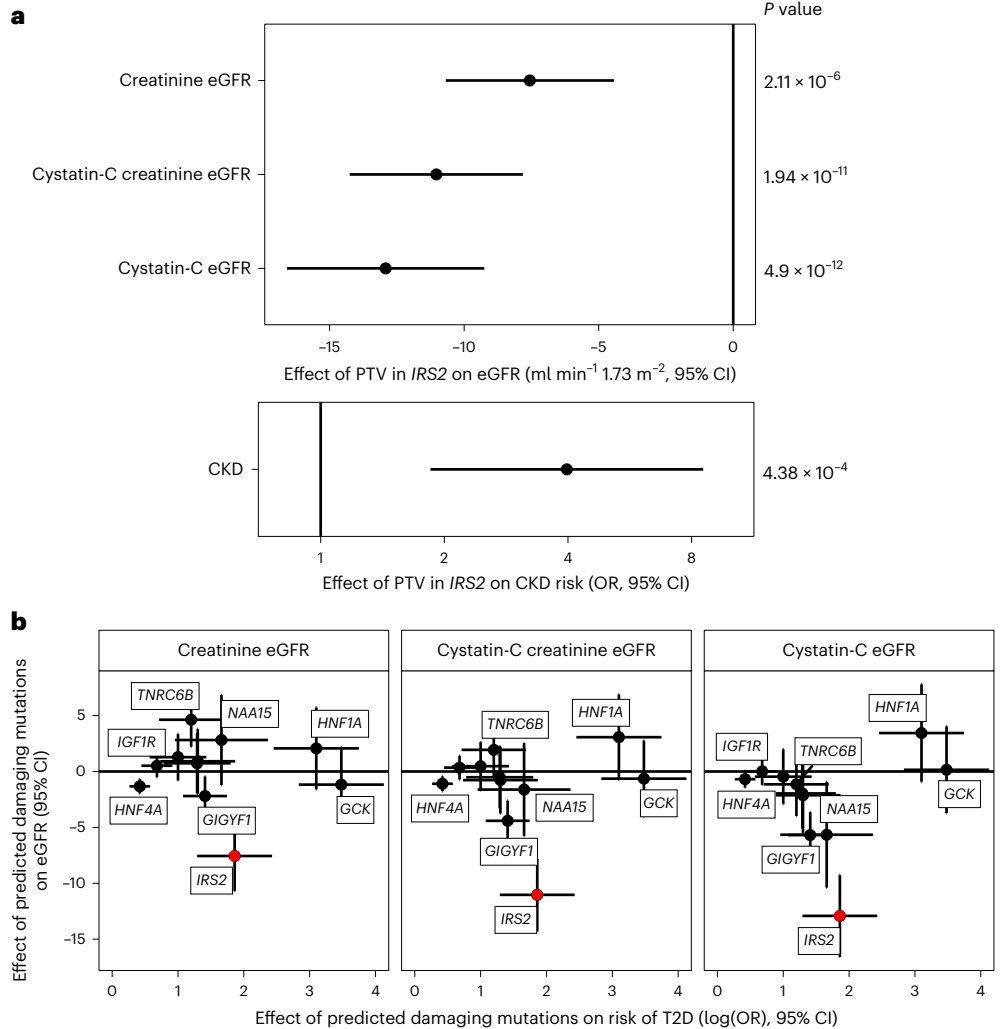

**Fig. 4 | Loss-of-function variants in *IRS2* increase CKD risk. a**, Effects of protein truncating variants in *IRS2* on various measures of eGFR (ml min$^{-1}$ 1.73m$^{-2}$) and CKD (OR) are plotted with 95% CIs. All *P* values are two-sided and unadjusted. The presented summary statistics are derived from linear (eGFR) and logistic regression (CKD risk) implemented in the GLM framework. **b**, Effects of rare predicted damaging mutations in the labeled genes on T2D risk are plotted (log(OR) T2D risk ± 95% CIs) against the effect on eGFR (beta estimate ± 95% CIs) across three different methods of estimation to illustrate that the effect of PTVs in *IRS2* on renal function seem independent of its effect on T2D. For clarity, only the Gene × Mask combination most significantly associated with T2D is plotted. All error bars represent 95% CIs. Plotted test statistics are derived from linear regression for eGFR and from logistic regression for T2D implemented using GLMs. $n_{\text{CKD, T2D}} = 489{,}941$; $n_{\text{Creatinine eGFR}} = 461{,}884$; $n_{\text{Cystatin-C eGFR}} = 462{,}081$; $n_{\text{Cystatin-C Creatinine eGFR}} = 461{,}543$.

be a secondary effect of increased adiposity (Supplementary Table 11). Although we did not observe any notable effects of *UBR3* PTVs on fat-free mass measurements, maximum hand-grip strength was nominally increased (Supplementary Table 11).

Alongside an increased risk of T2D, the risk of a clinical diagnosis of hypertension was increased significantly in carriers of *UBR2* and *UBR3* PTVs in our PheWAS analysis, but the effect observed in *UBR3* PTV carriers was nearly double that of *UBR2*. *UBR3* seems to increase adiposity from an early age—an effect not apparent for *UBR2* in UKBB. It is interesting to speculate that differential regulatory roles of these proteins throughout the life course may underlie the heterogeneity in their effects on cardiometabolic risk (Fig. 6).

## Discussion

By conducting genome-wide multi-ancestry gene-burden testing using WGS data from a cumulative total of >700,000 people, we identified several new BMI and T2D-associated genes. Compared with previous European-only analyses based on WES data, we increased carrier number and statistical power by incorporating all participants with available WGS data. We demonstrate that our findings from UKBB are

robust and reproducible, as several were replicated in an independent US population-based study (AoU) that has considerably different demographics, notably its younger age, higher baseline prevalence of T2D and enhanced ethnic diversity[36].

Our study also highlights some emerging challenges in conducting rare variant association studies across diverse populations. For example, we failed to replicate two gene masks using AoU data: *GCK* (Missense variants, REVEL >0.5) and *IGF1R* (REVEL >0.7). Both genes are robustly associated with T2D in UKBB, have >100 informative carriers in the AoU cohort and have a high probability of being true based on either known clinical associations with T2D (*GCK*) or orthogonal support from common-variant association studies (*IGF1R*)[11]. This may reflect specific challenges in the fidelity of missense classification tools across different pools of rare missense variants present in different cohorts with varying ethnic composition.

Our results provide several new biological insights into the determinants of human cardiometabolic health. The association with *RIF1*, a gene implicated in telomere regulation, DNA repair and replication timing, expands the list of DNA damage response genes involved in metabolic health[10]. The biological mechanisms behind these associations

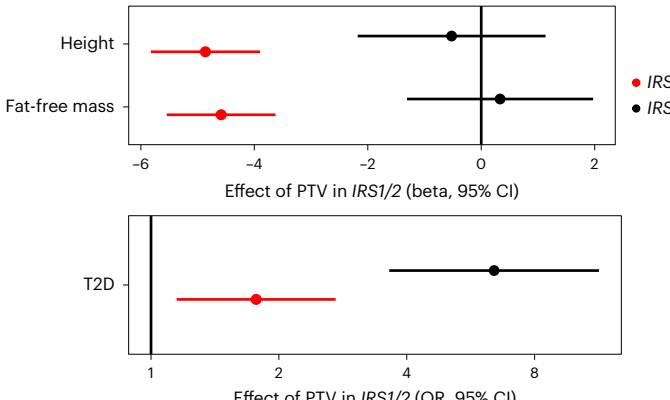

**Fig. 5 | Genetic evidence for functional heterogeneity of insulin receptor substrates in humans.** Effects of PTVs in *IRS1* and *IRS2* on continuous traits are beta-estimates from linear regression plotted in centimeters for height and kilograms for fat-free mass, and as OR from logistic regression for T2D. Odds of T2D are plotted on a log scale. All error bars represent 95% CIs. $n_{\text{Fat Free Mass}} = 481,100$; $n_{\text{Height}} = 488,455$; $n_{\text{T2D}} = 489,941$.

remain unclear. However, as the same variants in *RIF1* showed not even nominal association with recalled childhood adiposity in UKBB ($P > 0.05$), contrasting with their robust association with adult BMI ($P = 3.7 \times 10^{-10}$), we speculate that mechanisms that regulate neuronal degeneration might influence risk of adult-onset obesity[9].

We identified a robust, replicable association of PTVs in the critical signaling node in the insulin/IGF1 pathway, *IRS2* and T2D, with carriers exhibiting >3.6-fold increase in odds of diagnosis with T2D (OR 3.68 in AoU and 6.45 in UKBB). Although insulin resistance is well known as a necessary antecedent to the development of T2D, there is a longstanding interest in discerning the role of specific nodes in the insulin-signaling cascade in the development of insulin resistance and its related complications. By necessity, this work has been done largely in animal models, and the translational relevance of these findings to human health is uncertain. Candidate-gene testing and exome-sequencing studies of probands with extreme phenotypes identified in the clinical setting have been leveraged to provide insight into the function of several nodes of the insulin-signaling cascade in humans (*INSR*, *PIK3R1* and *AKT2*)[37–42]. Our findings definitively link *IRS2* to T2D as a component of the insulin-signaling cascade through study of rare loss-of-function variants using a hypothesis-free population-based sequencing approach. Using exome sequence data from a birth cohort, we identified two probands with normal fasting insulin levels in adolescence and early adulthood. These data makes it very unlikely that haploinsufficiency for IRS2 causes a highly penetrant monogenic severe insulin resistance syndrome. Rather, *IRS2* variants probably act as a risk modifier for T2D onset in later life by predisposing to insulin resistance and through effects in beta cells. To gain further insight into the specific phenotypic consequences of insulin/IGF1 resistance mediated by *IRS2* PTVs, we compared carriers of these variants with carriers of the other broadly expressed IRS protein, IRS1. We found that PTVs in *IRS1* conferred a much more modest effect on T2D risk compared to *IRS2*, but significantly reduced height and lean mass, phenotypes that were not associated with *IRS2* PTVs. These findings recapitulate observations first made in lower organisms, for example, *Irs1* knockout mice are insulin resistant and small but develop only modest dysglycemia due to compensatory changes in the beta cell[24,25]. In contrast, *Irs2* knockout mice grow normally and exhibit significant hepatic and skeletal muscle insulin resistance but, in contrast to their *Irs1* knockout counterparts, develop severe dysglycemia due to beta-cell failure[26]. Our results suggest that functional heterogeneity in IRS1/IRS2-mediated insulin/IGF1 signaling is conserved across species and is consistent with an important function of IRS2 in human beta-cell health, as has

been demonstrated in mice[43]. Future recall by genotype studies of *IRS2* PTV carriers with detailed assessment of glucose homeostasis and insulin sensitivity will be key in determining the relative contribution of insulin resistance and beta-cell failure to the development of T2D in the context of *IRS2* haploinsufficiency.

To gain a broader perspective of the effects of T2D risk and BMI-raising genes, we conducted a PheWAS for these genes. Notably, we observed that PTVs in *IRS2* significantly reduce eGFR independent of T2D status and cause a fourfold increase in the odds of CKD in UKBB. T2D risk genes did not generally increase CKD risk in UKBB, indicating that this is a specific effect of IRS2 disruption. Although the mechanistic basis of this association remains to be shown, insulin signaling exerts salutatory effects on podocyte health and function in mice[44–46], and germline loss of *Irs2* in mice results in smaller kidneys[47]. Both mechanisms could contribute to the adverse effects of PTVs in *IRS2*. Podocyte dysfunction and loss is a key early step in many forms of kidney disease, including diabetic nephropathy, and there is an increasing appreciation that the nephron number at birth (nephron endowment) is an important determinant of kidney health in later life[48,49], so both of these potential pathogenic mechanisms could be involved. Our demonstration of a causal role for *IRS2* in kidney health provides impetus to determine whether effects on renal health are mediated by a role of IRS2 in kidney development and nephrogenesis or by a regulatory role in postnatal renal physiology. If a renoprotective function of IRS2 in postnatal life exists, then examining the effects of risk factors for renal disease such as diabetes and obesity on IRS2-mediated signaling

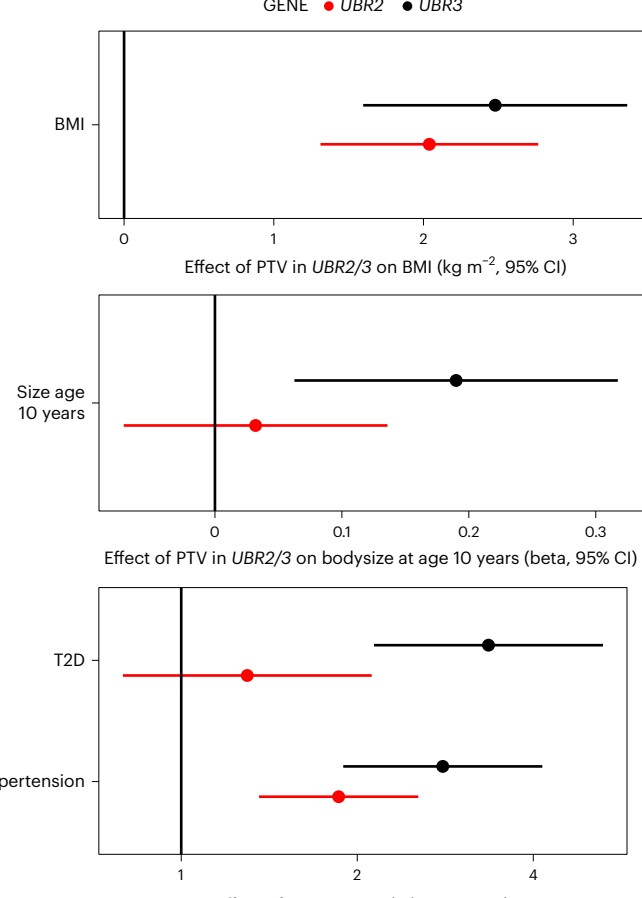

**Fig. 6 | Effects of PTVs in *UBR2* and *UBR3* on adiposity and cardiometabolic health.** Effects of PTVs in *UBR2* and *UBR3* on adiposity (adult BMI, body size age 10 years) and cardiometabolic outcomes are plotted. The points represent beta-estimates from linear regression for BMI (kg m$^{-2}$) and size age 10 years, and ORs derived from logistic regression for T2D and hypertension. All error bars represent 95% CIs. $n_{\text{BMI}} = 481,137$; $n_{\text{size age 10}} = 479,615$; $n_{\text{T2D, hypertension}} = 489,941$.

could highlight a new and potentially modifiable mechanism of kidney disease. Our findings support the notion that *IRS2* PTVs cause CKD independent of chronic hyperglycemia as we found no similar associations for several genes with stronger effects on T2D risk, including *HNF1A*, where microvascular complications are known to develop if glycemic control is suboptimal.

An intriguing finding was the association of the related genes *UBR2* and *UBR3* with BMI, with the latter also elevating T2D risk in a manner only partially dependent on its effect on BMI. Both genes encode E3-ubiquitin ligases: UBR2 functions as an effector of the N-degron pathway recognizing modified N-terminal amino acids and targeting their host protein for degradation, whereas UBR3, despite structural homology to UBR2, lacks canonical N-recognin activity[30,31]. Although both UBR2 and UBR3 are relatively broadly expressed, UBR3 is enriched in a number of sensory tissues, including tongue, ear and olfactory epithelia, which may have relevance to its effects on BMI[17]. Both UBR2 and UBR3 are relatively enriched in expression in skeletal muscle. Although the specific effects of these proteins in muscle are unclear, UBR3 may play a nonredundant role in skeletal muscle function as carriers of PTVs in *UBR3* had reductions in grip strength. Our work clearly highlights UBR2 and UBR3 as important regulators of cardiometabolic health, but further study exploring their substrates and function are necessary to gain a mechanistic understanding of their effects on BMI and T2D.

There are translational implications of our findings. Notably, in a PheWAS of potentially relevant traits, we observed a strong association between predicted damaging missense mutations in *HNF4A* and reduced circulating SHBG. Although it has been noted that HNF4A can activate the SHBG promoter[23], and a causal relationship between HNF4A and circulating SHBG has been suggested[50], our study provides genetic evidence in humans which supports this notion. Pathogenic mutations in *HNF4A* cause a type of monogenic diabetes onset of the young (MODY); we speculate that people with apparent T2D and low SHBG without significant insulin resistance may be enriched for *HNF4A* mutations. We have also identified phenotypic consequences of loss of *IRS2* in humans. Our work provides an impetus for research-based genetic testing of people exhibiting cases of atypical diabetes[51], particularly if they also have CKD and/or a monogenic cause is suspected. In summary, our study expands the number of genes directly implicated in metabolic health by rare human genetic variation and further illustrates the benefit of genome over exome sequencing for the discovery of rare variants associated with disease.

## Online content

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

## Methods

### Ethics

Our research complies with all relevant ethical regulations. All studies included in this research were approved by the relevant board or committee. UKBB has approval from the North West Multicentre Research Ethics Committee (REC reference 13/NW/0157) as a Research Tissue Bank (RTB) approval, and informed consent was provided by each participant. This approval means that researchers do not require separate ethical clearance and can operate under RTB approval. This RTB approval was granted initially in 2011 and is renewed every 5 years; hence, UKBB successfully renewed approval in 2016 and 2021. All work in UKBB reported in this manuscript was performed under UKBB application numbers 20361 and 52293. The collection of participant information adhered to the AoU Research Program Operational Protocol (https://allofus.nih.gov/article/all-us-research-program-protocol). The AoU Institutional Review Board (IRB) (https://allofus.nih.gov/about/who-we-are/institutional-review-board-irb-of-all-of-us) is charged with reviewing the protocol, informed consent and other participant-facing materials for the AoU Research Program. The IRB follows the regulations and guidance of the Office for Human Research Protections for all studies, ensuring that the rights and welfare of research participants are overseen and protected uniformly. For ALSPAC, ethical approval for the study was obtained from the ALSPAC Ethics and Law Committee and the Local Research Ethics Committees.

### UKBB WGS data processing

The WGS of UKBB participants is described in detail in ref. 7. In brief, 490,640 UKBB participants were sequenced to an average depth of 32.5× using Illumina NovaSeq 6000 platform. Variants were jointly called using Graphtyper[52], which resulted in 1,037,556,156 and 101,188,713 high-quality (AAscore <0.5 and <5 duplicate inconsistencies) single-nucleotide polymorphisms (SNPs) and indels, respectively.

We further processed the jointly called genotype data in Hail v.0.2[53], where multi-allelic sites were first split and normalized. Variants were then filtered based on low allelic balance (ABHet <0.175, ABHom <0.9), low quality-by-depth (QD) normalized score (QD < 6), low phred-scaled quality score (QUAL < 10) and high missingness (call rate <90%). For the analysis in the European-ancestry cohort (see below), we further removed variants that failed test for Hardy–Weinberg equilibrium ($P < 1 \times 10^{-100}$) within this cohort.

Variants were annotated using Ensembl variant effect predictor (VEP)[54] v.108.2 with the LOFTEE plugin[55]. Combined annotation-dependent depletion (CADD) annotations were based on precomputed CADD[56] v.1.7 annotations for all SNPs and gnomAD v.4 indels. REVEL[15] annotations were obtained from the 3 May 2021 release of precomputed REVEL scores for all SNPs. We prioritized the individual consequence for each variant based on severity, which was defined by VEP. The PTV category is the combination of stop-gained, frameshift, splice acceptor and splice donor variants. The missense and synonymous variants were adopted directly from VEP. Only the variants on autosomes and chromosome X, which were within ENSEMBL protein-coding transcripts, were included in our downstream analysis.

Demographics of the study population are presented in Supplementary Table 16.

### European ancestry definition in UKBB WGS

We defined a European-ancestry cohort as people who most resembled the NFE (non-Finnish European) population as labeled in the gnomAD v.3.1 dataset[55]. This NFE group was one of nine ancestry groups labeled in gnomAD, which was based on Human Genome Diversity Project and 1000 Genomes Project samples. Variant loadings for 76,399 high-quality informative variants from gnomAD were used to project the first 16 principal components onto all UKBB WGS samples. A random forest classifier trained on the nine ancestry labels in gnomAD was then used to calculate probabilities that reflect the similarity between the UKBB participant and each of the gnomAD ancestry labels.

### Phenotype preparation in UKBB

Binary outcomes were prepared using a combination of hospital episode statistics (UKBB showcase IDs: 41202, 41204, 41200, 41210) primary care records (UKBB showcase IDs: 42040), death certificates (UKBB showcase IDs: 40001, 40002) and self-reported medical conditions (UKBB showcase ID: 20002). Qualifying codes pertaining to each condition are listed in Supplementary Table 10. Any participant with a qualifying code was considered a case, those without a qualifying code were considered controls. For T2D, qualifying terms included codes specifying diagnoses of noninsulin-dependent diabetes, T2D, and insulin-treated T2D. Participants who self-reported a history of T2D were also classified as cases. For CKD, diagnostic codes included those specifying chronic renal failure, chronic renal impairment, CKD, end-stage renal failure, hypertensive renal disease with renal failure, or codes indicative of preparation or receipt of renal replacement therapy. Participants who self-reported renal/kidney failure, dialysis or procedures to prepare for peritoneal or hemodialysis were specified as cases. For T2D and CKD phenotype definitions, all participants who did not meet the qualifying terms were classified as controls (see Supplementary Table 10 for a full list of qualifying codes). Thinness was defined as having the lowest 5% of BMI. Metabolic dysfunction-associated steatotic liver disease required the presence of steatosis and a qualifying metabolic risk factor, namely obesity, T2D or other metabolic dysregulation[57]. Steatosis was defined using the fatty liver index, a composite measurement of triglycerides, glutamyl-transferase, waist circumference and BMI that ranges from 0 to 100 (ref. 58). Specific fatty liver index cut-offs according to participant sex, BMI and waist circumference as described in ref. 59 were applied to define presence of steatosis.

### Genome-wide gene-burden testing in the UKBB

BOLT-LMM[60] v.2.4.1 was used as our primary analytical software to conduct gene-burden tests.

To run BOLT-LMM, we first derived a set of genotypes consisting of common (MAF >0.01) linkage disequilibrium (LD)-pruned (LD $r^2$ < 0.1) variants in people with WGS data to build the null model. Pruning was conducted using PLINK2[61] on a random subset of 50,000 individuals (options in effect: −maf 0.01 −thin-indiv-count 50,000 −indep-pairwise 1,000 kb 0.1).

We adopted the same strategies used in our previous analyses using WES data[9,11]. We generate the dummy genotype files in which each gene-mask combination was represented by a single variant, which were required as the genotype input for BOLT-LMM. We then coded people with a qualifying variant within a gene as heterozygous, regardless of the total number of variants they carried in that gene. We then created the dummy genotypes for the MAF <0.1% high-confidence PTVs as defined by LOFTEE, missense variants with REVEL >0.5 and missense variants with REVEL >0.7. After getting all required inputs, BOLT-LMM was used to analyze BMI and T2D using default parameters except for the inclusion of the 'lmmInfOnly' flag. The covariates included in our analysis are age, age$^2$, sex, age × sex, the first 20 principal components as calculated from all WGS samples and the WGS-released batch (Vanguard project, Sanger: 49,932, Sanger: 193,075, deCode: 247,504). Different from our previous studies, we included all samples without restricting their ancestries to maximize the sample size. Only people who withdrew consent or had missing phenotypes and covariates were excluded; filtering resulted in 481,137 and 489,941 samples remaining for BMI and T2D, respectively.

To identify single variants driving a given association within a single gene, we performed a leave-one-out analysis for all identified genes using a generalized linear model (GLM) in R v.4.0.2 by dropping the variants contained in the gene-mask combination one at a time.

To test whether our significant burden test results are independent of common-variant GWAS associations, we generated polygenic risk scores for each trait and included these as covariates in our linear mixed model. Independent genome-wide significant ($P < 5 \times 10^{-8}$) variants from existing single-variant GWAS summary statistics for each trait were first identified using GCTA-COJO[62]. Polygenic risk scores in each UKBB participant were then calculated as the weighted sum of the person's genotypes across the significant single variants, where weights were derived from the variants' beta coefficients in the corresponding GWAS. This score was then included as an additional covariate in the burden analysis as implemented in BOLT-LMM described above. As BOLT-LMM use a linear mixed model, we estimated and reported the OR using the generalized linear model in R v.4.0.2 for all T2D-associated genes.

In an additional analysis designed to exclude that our new, replicated rare variant associations were the result of confounding by LD with common variants we interrogated marker level results from WGS-analyses of BMI and T2D. Regional common variants that could conceivably be driving the rare variant associations (MAF >0.001, $P < 6.15 \times 10^{-7}$, ± 500 kb from index gene) were extracted and clumped ($r^2 < 0.001$) to identify approximately independent variants, which were then included as covariates in a generalized linear model with the cognate gene-burden mask as the predictor variable of interest. As in our discovery analysis, age, age$^2$, sex, age × sex and the first 20 principal components as calculated from all WGS samples, and the WGS-released batch were included as covariates.

### Replication in AoU study

Participants analyzed in this study were selected from the AoU Research Program cohort[36]. The collection of participant information adhered to the AoU Research Program Operational Protocol (https://allofus.nih.gov/article/all-us-research-program-protocol). Detailed methodologies regarding genotyping, ancestry classification, quality control measures and the methodology for excluding related participants are thoroughly documented in the AoU Research Program Genomic Research Data Quality Report (https://support.researchallofus.org/hc/en-us/articles/4617899955092-All-of-Us-Genomic-Quality-Report).

We conducted our analysis on short-read WGS data (v.7.1) subsetted to the protein-coding exome, focusing on two phenotypes: BMI and T2D. The analysis encompassed 219,015 unrelated people, including 112,526 of European ancestry, 46,414 of African/African American ancestry, 34,865 of American Admixed/Latino and 25,210 various other ancestries (see Supplementary Table 3 for detailed sample size information). Ancestry assignment was conducted centrally by AoU. Briefly, a random forest classifier was trained on data from the Human Genetic Diversity Project and 1000 Genomes Project. This classifier was then applied to the AoU data. Further information is available from the AoU (https://support.researchallofus.org/hc/en-us/articles/4617899955092-All-of-Us-Genomic-Quality-Report-ARCHIVED-C2022Q4R9-CDR-v7).

BMI data were derived from the 'body mass index (BMI) [Ratio]' metric (Concept Id 3038553) within the 'Labs and Measurements' domain. BMI values <10 or >100 were excluded and the earliest remaining value recorded and corresponding age was used. The 'Type 2 diabetes mellitus' identifier (Concept Id 201826, https://databrowser.researchallofus.org/ehr/conditions/201826) in the 'Conditions' domain facilitated the identification of T2D cases, and the age corresponding to the earliest diagnosis of T2D was used. The participants' ages were calculated by subtracting the birth year from the timestamp of the earliest record. Among these people, 32,462 were identified as T2D cases, and 186,553 served as controls. Only people aged over 18 years were included in the analyses. Only a small proportion of episodes that indicated a diagnosis of T2D had a contemporaneous BMI measurement. As such, to adjust T2D for BMI, we used two approaches: the median BMI value recorded was included as a covariate in the model

or the BMI record closest to T2D was used. Population demographics by ancestry are described in Supplementary Table 17.

Gene-based burden tests were applied to variants with MAF <0.001 that met prespecified bioinformatic criteria and were in selected genes (for example, those significant in UKBB discovery). Note that, due to different population composition, variant MAF will differ between AoU and UKBB. Burden tests were conducted using STAAR (variant-set test for association using annotation information)[63] implemented in STAARpipeline[64] (R package v.0.9.7), with covariates adjustments for age, age$^2$, sex, age × sex, and the first 16 principal components. The criteria for gene-burden masks followed the methodology of the main UKB analyses.

### Power calculations

To estimate statistical power for replication in the AoU study, we first corrected effect estimates in the discovery analysis for winners' curse using the bootstrap method[65] implemented in the winner's curse package in R (https://amandaforde.github.io/winnerscurse/). For T2D, the resulting effect estimates (betas from a linear mixed model) were transformed to odds ratios[66] (https://shiny.cnsgenomics.com/LMOR/). Power calculations using the relevant winners' curse corrected effect estimates were then conducted in genpwr (https://cran.r-project.org/web/packages/genpwr/vignettes/vignette.html).

### UKBB WES processing

To quantify the gain from WGS versus WES in UKBB, we compared variant counts between our WGS data with those from the 450,000 original quality functional equivalence release of the UKBB WES data (454,756 participants total). We processed multisample pVCFs using Hail[53] v.0.2, where multi-allelic sites were first split and normalized. Sites were then excluded if they failed the following quality metrics: for SNPs, ABHet <0.175, QD <2, QUAL <30, SOR >30, FS >60, MQ <40, MQRankSum <−12.5 and ReadPosRankSum <−8; for indels: ABHet <0.175, QD <2, QUAL <30, FS >200 and ReadPosRankSum <−20, resulting in 23,273,514 variants available for analysis. People with high heterozygosity rates, discordant WES genotypes compared to array and discordant reported versus genetic sex were removed, resulting in 453,931 participants. Variants were annotated using the identical VEP pipeline, LOFTEE, CADD and REVEL annotations as described for WGS.

### PheWAS of identified BMI-associated and T2D-associated genes in UKBB

We ran association tests between each identified genes carriers and a list of representative phenotypes (full list can be found in Supplementary Tables 10 and 11) available in the UKBB using R v.4.0.2 including the same covariates we used in our genome-wide gene-burden tests. We also extracted the phenotypic associations with $P < 0.05$ for all genes we identified in our analysis from AstraZeneca PheWAS Portal[67] (version: UKBB 470 K WES v.5; Supplementary Tables 18 and 19).

### BMI and T2D GWAS lookup

Identified genes were queried for proximal BMI and T2D GWAS signals, using data from the largest published GWAS meta-analyses. For BMI, we used data from the GIANT consortium[68], which includes data on up to 806,834 individuals. For T2D, we used data from the DIAGRAM consortium[69], which included up to 428,452 T2D cases and 2,107,149 controls.

For each of these GWAS, we performed signal selection and prioritized causal GWAS genes using the 'GWAS to Genes' pipeline as described elsewhere[18]. The genes identified previously were annotated if their start or end sites were within 500 kb up- or downstream of GWAS signals in the two meta-analyses, using the National Center for Biotechnology Information RefSeq gene map for GRCh37, and overlayed with further supporting functional dataset information. For further details about the specific application of this method, see ref. 18.

## Assessment for severe insulin resistance in carriers of *IRS2* PTVs in a UK birth cohort

ALSPAC is a prospective birth cohort from the southwest of England that recruited >75% of all pregnancies delivered in the Greater Bristol area between 1990 and 1992 (refs. 70–73). The study has currently enrolled 14,833 unique women (G0 mothers), 3,807 G0 partners and 14,901 children. Full details of the cohort and study design are available at http://www.alspac.bris.ac.uk. Please note that the study website contains details of all the data that is available through a fully searchable data dictionary and variable search tool (http://www.bristol.ac.uk/alspac/researchers/our-data/). Exome sequencing data from 8,605 children and 3,389 of their parents was interrogated for carriers of any high-confidence protein truncating variants in *IRS2* as defined by LOFTEE with MAF <1%. Two such carriers were identified (Supplementary Table 14), and insulin and glucose measurements extracted from available data and insulin levels were compared to population specific reference ranges (2.5th and 97.5th centile) and commonly used clinical cut-offs for severe insulin resistance (<150 pmol l$^{-1}$). Insulin was measured using either an ELISA (Mercodia) or an ECLIA (Roche)[74]. Details of exome sequencing, quality control, variant calling and annotation have been described in https://wellcomeopenresearch.org/articles/9-390/v1.

## Lookup of effects of *IRS2* PTVs on CKD-related traits in the AoU cohort

To provide supporting evidence of an effect of loss-of-function variants in *IRS2* on CKD, we leveraged the results of a recent PheWAS conducted in the AoU cohort, publicly accessible using the 'All by All web browser' (https://allbyall.researchallofus.org/). The relevant gene page is available directly at https://allbyall.researchallofus.org/app?state=%7B%22regionId%22%3Anull%2C%22geneId%22%3A%22ENSG00000185950%22%2C%22resultIndex%22%3A%22gene-phewas%22%2C%22resultLayout%22%3A%22full%22%2C%22analysisId%22%3A%223027114%22%2C%22variantId%22%3Anull%2C%22burdenSet%22%3A%22pLoF%22%2C%22ancestryGroup%22%3A%22meta%22%2C%22phewasOpts%22%3Atrue%2C%22selectedContig%22%3A%22all%22%2C%22hideGeneOpts%22%3Afalse%7D.

### Reporting summary

Further information on research design is available in the Nature Portfolio Reporting Summary linked to this article.

## Data availability

The UKBB phenotype, WEG and WES data described here are publicly available to registered researchers through the UKB data access protocol. Information about registration for access to the data is available at: https://www.ukbiobank.ac.uk/enable-your-research/apply-for-access. Data for this study were obtained under Resource Application nos. 20361 and 68574. The AoU phenotype and WGS data described here are available to registered researchers through the AoU data access protocol. Information about registration for access to the data is available at https://www.researchallofus.org/register/. Summary statistics from the exome-wide association studies are reported in the Supplementary Tables. All bona fide researchers can apply to use ALSPAC data for health-related research that is in the public interest. Information regarding the ALSPAC cohort and data access is available at: https://www.bristol.ac.uk/alspac/researchers/our-data/.

## Code availability

All analyses were performed used publicly available software. No custom code was developed.

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

## Acknowledgements

We thank the participants and investigators in the UKBB study (resource application nos. 20361 and 52293) and AoU study who made this work possible. We are extremely grateful to all the families who took part in ALSPAC, the midwives for their help in recruiting them, and the whole ALSPAC team, which includes interviewers, computer and laboratory technicians, clerical workers, research scientists, volunteers, managers, receptionists and nurses. S.O.'R. is supported by a Wellcome Investigator Award (WT 214274/Z/18/Z) and Cambridge NIHR Biomedical Research Centre. S.L. was funded by a Wellcome Trust Clinical PhD Fellowship (WT 225479/Z/22) and by an Academic Clinical Lectureship from Queen's University Belfast and the Northern Ireland Department of Health. Y.Z., E.J.G., K.A.K., J.R.B.P. and K.K.O. acknowledge funding from the Medical Research Council (Unit Program: MC_UU_00006/2). Y.Z. is supported by Changping Laboratory. X.L. is supported by the research start-up funds from the Department of Biostatistics and the Department of Genetics at the University of North Carolina at Chapel Hill. The UK Medical Research Council and Wellcome (grant ref: 217065/Z/19/Z) and the University of Bristol provide core support for ALSPAC. A comprehensive list of grants funding is available on the ALSPAC website (http://www.bristol.ac.uk/alspac/external/documents/grant-acknowledgements.pdf). This research was supported in part by the Intramural Research Program of the National Institutes of Health (NIH). The contributions of the NIH authors were made as part of their official duties as NIH federal employees, are in compliance with agency policy requirements, and are considered Works of the United States Government. However, the findings and conclusions presented in this paper are those of the authors and do not necessarily reflect the views of the NIH or the U.S. Department of Health and Human Services. For the purpose of open access, the authors have applied a CC BY public copyright license to any author accepted manuscript version arising from this submission.

## Author contributions

Y.Z., S.L., D.B.S., C.B.-D, K.K.O., H.Z., R.S., S.O.'R. and J.R.B.P. designed the study and/or contributed to biological interpretation of findings. Y.L., J.D., Y.Z., S.L., J.L., A.C., E.J.G. and K.A.K. supported genetic analyses and genotype–phenotype association testing of the UKBB data. X.L., X.H. and H.Z. performed replication analyses in the AoU study, and M.C.-G., L.F., K.H. and N.T. performed analyses related to *IRS2* PTV carriers in ALSPAC. All authors reviewed and contributed toward the drafting of the manuscript.

## Competing interests

J.L., A.C., Y.L., J.D., C.B.-D. and R.S. are employees and stockholders of GSK. J.R.B.P. and E.J.G. are employees and shareholders of Insmed. J.R.B.P. receives research funding from GSK. Y.Z. was a UK University worker at GSK during this work. S.O.'R. has undertaken remunerated consultancy work for Pfizer, Third Rock Ventures, AstraZeneca, NorthSea Therapeutics and Courage Therapeutics and is a scientific founder of Marea Therapeutics. S.L. performs paid consultancy for Eolas Medical. The other authors declare no competing interests.

## Additional information

**Correspondence and requests for materials** should be addressed to John R. B. Perry.

# Reporting Summary

## Statistics

For all statistical analyses, confirm that the following items are present in the figure legend, table legend, main text, or Methods section.

| n/a | Confirmed | |
|---|---|---|
| ☐ | ☒ | The exact sample size ($n$) for each experimental group/condition, given as a discrete number and unit of measurement |
| ☒ | ☐ | A statement on whether measurements were taken from distinct samples or whether the same sample was measured repeatedly |
| ☐ | ☒ | The statistical test(s) used AND whether they are one- or two-sided *Only common tests should be described solely by name; describe more complex techniques in the Methods section.* |
| ☐ | ☒ | A description of all covariates tested |
| ☐ | ☒ | A description of any assumptions or corrections, such as tests of normality and adjustment for multiple comparisons |
| ☐ | ☒ | A full description of the statistical parameters including central tendency (e.g. means) or other basic estimates (e.g. regression coefficient) AND variation (e.g. standard deviation) or associated estimates of uncertainty (e.g. confidence intervals) |
| ☐ | ☒ | For null hypothesis testing, the test statistic (e.g. $F$, $t$, $r$) with confidence intervals, effect sizes, degrees of freedom and $P$ value noted *Give P values as exact values whenever suitable.* |
| ☒ | ☐ | For Bayesian analysis, information on the choice of priors and Markov chain Monte Carlo settings |
| ☒ | ☐ | For hierarchical and complex designs, identification of the appropriate level for tests and full reporting of outcomes |
| ☒ | ☐ | Estimates of effect sizes (e.g. Cohen's $d$, Pearson's $r$), indicating how they were calculated |

*Our web collection on statistics for biologists contains articles on many of the points above.*

## Software and code

Policy information about availability of computer code

| Data collection | N/A |
|---|---|
| Data analysis | Graphtyper, Hail v0.2, Ensembl Variant Effect Predictor (VEP) v108.2, BOLT-LMM v2.4.1, R v4.0.2, STAARpipeline v0.9.7, winnerscurse v0.1.1, genpwr 1.0.4, |

For manuscripts utilizing custom algorithms or software that are central to the research but not yet described in published literature, software must be made available to editors and reviewers. We strongly encourage code deposition in a community repository (e.g. GitHub). See the Nature Portfolio guidelines for submitting code & software for further information.

## Data

Policy information about availability of data

All manuscripts must include a data availability statement. This statement should provide the following information, where applicable:
- Accession codes, unique identifiers, or web links for publicly available datasets
- A description of any restrictions on data availability
- For clinical datasets or third party data, please ensure that the statement adheres to our policy

The UK Biobank phenotype, whole-genome and whole-exome sequencing data described here are publicly available to registered researchers through the UKB data access protocol. Information about registration for access to the data is available at: https://www.ukbiobank.ac.uk/enable-your-research/apply-for-access.  Data for this study were obtained under Resource Application Numbers: 20361 and 68574.

# Research involving human participants, their data, or biological material

Policy information about studies with human participants or human data. See also policy information about sex, gender (identity/presentation), and sexual orientation and race, ethnicity and racism.

| Reporting on sex and gender | In our analyses, we included both males and females and we adjusted sex in our regression analysis. |
|---|---|

| Reporting on race, ethnicity, or other socially relevant groupings | In both UK Biobank and All of Us Research Program, we include all samples without restricting their genetic ancestry in our primary analysis. In UK Biobank, to make comparison, we performed the European only analysis. We defined a subset of European ancestry samples using a random forest classifier approach. |
|---|---|

| Population characteristics | The UK Biobank is a large prospective cohort that recruited approximately 500,000 participants aged 40 to 69 years across the island of Great Britain. A broad range of phenotypic and health-related information was collected from each participant, including physical measurements, lifestyle indicators, biomarkers in blood and urine, imaging, and routine health record data. See Supplementary Tables 16 for additional demographic data.

The All of Us Research Program is a longitudinal cohort study aiming to enrol a diverse group of at least one million individuals aged from 18 years across the USA to accelerate biomedical research and improve human health. Participant data include a rich combination of phenotypic and genomic data. Participants are asked to complete consent for research use of data, sharing of electronic health records (EHRs), donation of biospecimens (blood or saliva, and urine), in-person provision of physical measurements (height, weight and blood pressure) and surveys initially covering demographics, lifestyle and overall health. Participants are also consented for recontact. See supplementary table 17 for additional summary information on population demographics.

ALSPAC is a prospective birth cohort from the southwest of England that recruited >75% of all pregnancies delivered in the Greater Bristol area between 1990 and 199271–74. The study has currently enrolled 14,833 unique women (G0 mothers), 3,807 G0 partners and 14,901 children. Full details of the cohort and study design are available at http://www.alspac.bris.ac.uk. Please note that the study website contains details of all the data that is available through a fully searchable data dictionary and variable search tool (http://www.bristol.ac.uk/alspac/researchers/our-data/). Exome sequencing data from 8,605 children and 3,389 of their parents was interrogated for carriers of any high confidence protein truncating variants in IRS2 as defined by LOFTEE with MAF<1%. 2 such carriers were identified (Supplementary table 13) and insulin and glucose measurements extract from available data and insulin levels compared to population specific reference ranges (2.5th and 97.5th centile) and commonly used clinical cut-offs for severe insulin resistance. (<150 pmol/L). Insulin was measured using either an ELISA (Mercodia) or an ECLIA (Roche)75. Details of exome sequencing, quality control, variant calling and annotation have been described previously [https://wellcomeopenresearch.org/articles/9-390/v1]. Ethical approval for the study was obtained from the ALSPAC Ethics and Law Committee and the Local Research Ethics Committees. |
|---|---|

| Recruitment | Participants of the UK Biobank aged from 40 to 69, who were registered with NHS and living up to about 25 miles from one of the 22 study assessment centres were invited to participate in 2006-2010.

The All of Us Research Program seeks to recruit persons in demographic categories that have been and continue to be underrepresented in biomedical research; such persons typically have relatively poor access to good health care. Participants enroll digitally through the All of Us website (https://joinallofus.org) or a smartphone app.

ALSPAC is a prospective birth cohort from the southwest of England that recruited >75% of all pregnancies delivered in the Greater Bristol area between 1990 and 199271–74. |
|---|---|

| Ethics oversight | The UK Biobank has approval from the North West Multi-centre Research Ethics Committee (REC reference 13/NW/0157, https://www.ukbiobank.ac.uk/media/lcvbdoik/21-nw-0157-favourable-opinion-with-conditions-18-06-2021.pdf) as a Research Tissue Bank (RTB) approval and informed consent (https://www.ukbiobank.ac.uk/media/t22hbo35/consent-form.pdf) was provided by each participant. This approval means that researchers do not require separate ethical clearance and can operate under the RTB approval. This RTB approval was granted initially in 2011 and it is a renewal on a 5-yearly cycle; hence UK Biobank successfully applied to renew it in 2016 and 2021.

For All of Us, informed consent for all participants is conducted in person or through an eConsent platform that includes primary consent, HIPAA Authorization for Research use of EHRs and other external health data, and Consent for Return of Genomic Results. The protocol was reviewed by the Institutional Review Board (IRB) of the All of Us Research Program. The All of Us IRB follows the regulations and guidance of the NIH Office for Human Research Protections for all studies, ensuring that the rights and welfare of research participants are overseen and protected uniformly.

Ethical approval for the work related to the ALSPAC study was obtained from the ALSPAC Ethics and Law Committee and the Local Research Ethics Committees. |
|---|---|

Note that full information on the approval of the study protocol must also be provided in the manuscript.

# Field-specific reporting

Please select the one below that is the best fit for your research. If you are not sure, read the appropriate sections before making your selection.

☒ Life sciences ☐ Behavioural & social sciences ☐ Ecological, evolutionary & environmental sciences

For a reference copy of the document with all sections, see nature.com/documents/nr-reporting-summary-flat.pdf

# Life sciences study design

All studies must disclose on these points even when the disclosure is negative.

| | |
|---|---|
| Sample size | We used the full available sample with whole-genome sequencing data in UK Biobank (N=490,640) for discovery analyses. This is a convenience sample using all available data in one of the worlds largest population genetics resources with whole genome sequencing information. |
| Data exclusions | Only individuals with missing phenotype or covariates were excluded from analysis. This decision was made prior to performing any downstream analysis. |
| Replication | We replicated findings in All of Us Research Program (total N=219,015). All attempted replication has been reported in the manuscript without exception. |
| Randomization | N/A |
| Blinding | Blinding was not applicable. This study involved a retrospective analysis of genetic and phenotypic data from the UK Biobank. All genotyping and phenotype collection were completed prior to analysis, and no experimental interventions were performed. Statistical analyses (e.g., association testing) were conducted using predefined models without manual interpretation or subjective outcome assessment. |

# Reporting for specific materials, systems and methods

We require information from authors about some types of materials, experimental systems and methods used in many studies. Here, indicate whether each material, system or method listed is relevant to your study. If you are not sure if a list item applies to your research, read the appropriate section before selecting a response.

## Materials & experimental systems

| n/a | Involved in the study |
|---|---|
| ☒ ☐ | Antibodies |
| ☒ ☐ | Eukaryotic cell lines |
| ☒ ☐ | Palaeontology and archaeology |
| ☒ ☐ | Animals and other organisms |
| ☒ ☐ | Clinical data |
| ☒ ☐ | Dual use research of concern |
| ☒ ☐ | Plants |

## Methods

| n/a | Involved in the study |
|---|---|
| ☒ ☐ | ChIP-seq |
| ☒ ☐ | Flow cytometry |
| ☒ ☐ | MRI-based neuroimaging |

# Plants

| | |
|---|---|
| Seed stocks | *Report on the source of all seed stocks or other plant material used. If applicable, state the seed stock centre and catalogue number. If plant specimens were collected from the field, describe the collection location, date and sampling procedures.* |
| Novel plant genotypes | *Describe the methods by which all novel plant genotypes were produced. This includes those generated by transgenic approaches, gene editing, chemical/radiation-based mutagenesis and hybridization. For transgenic lines, describe the transformation method, the number of independent lines analyzed and the generation upon which experiments were performed. For gene-edited lines, describe the editor used, the endogenous sequence targeted for editing, the targeting guide RNA sequence (if applicable) and how the editor was applied.* |
| Authentication | *Describe any authentication procedures for each seed stock used or novel genotype generated. Describe any experiments used to assess the effect of a mutation and, where applicable, how potential secondary effects (e.g. second site T-DNA insertions, mosiacism, off-target gene editing) were examined.* |

