## [Peer Review File · Nature Genetics]

Population-scale gene-based analysis of whole genome sequencing provides novel insights into metabolic health

Corresponding Author: Professor John Perry

Version 0:

Decision Letter:

22nd July 2024

Dear John,

Your Article "Population scale whole genome sequencing provides novel insights into cardiometabolic health" has been seen by four referees. (Reviewers #1 and #2 assessed the paper together and provided a single joint review.) You will see from their comments below that, while they find your work of interest, they have raised substantial points that must be addressed. In light of these comments, we cannot accept the manuscript for publication at this time, but we would be interested in considering a suitably revised version that addresses the referees' concerns.

We hope you will find the referees' comments useful as you decide how to proceed. If you wish to submit a substantially revised manuscript, please bear in mind that we will be reluctant to approach the referees again in the absence of major revisions.

To guide the scope of the revisions, the editors discuss the referee reports in detail within the team, including with the chief editor, with a view to identifying key priorities that should be addressed in revision and sometimes overruling referee requests that are deemed beyond the scope of the current study. In this case, we ask that you address all technical queries related to the association analyses and their interpretation, revising claims where needed, and extend the analyses to provide further insights into the relative performance of whole genome sequencing-based vs. alternate strategies. We hope you will find this prioritized set of referee points to be useful when revising your study. Please do not hesitate to get in touch if you would like to discuss these issues further.

If you choose to revise your manuscript taking into account all reviewer and editor comments, please highlight all changes in the manuscript text file. At this stage, we will need you to upload a copy of the manuscript in MS Word .docx or similar editable format.

*2) If you have not done so already, please begin to revise your manuscript so that it conforms to our Article format instructions, available [here](http://www.nature.com/ng/authors/article_types/index.html). Refer also to any guidelines provided in this letter.

*3) Include a revised version of your Reporting Summary: <https://www.nature.com/documents/nr-reporting-summary.pdf> It will be available to referees (and, potentially, statisticians) to aid in their evaluation if the manuscript goes back for peer review.

Link Redacted

If you wish to submit a suitably revised manuscript, we hope to receive it within 3-6 months. If you cannot send it within this time, please let us know. We will be happy to consider your revision so long as nothing similar has been accepted for publication at Nature Genetics or published elsewhere. Should your manuscript be substantially delayed without notifying us in advance and your article is eventually published, the received date would be that of the revised, not the original, version.

Nature Genetics is committed to improving transparency in authorship. As part of our efforts in this direction, we are now requesting that all authors identified as 'corresponding author' on published papers create and link their Open Researcher and Contributor Identifier (ORCID) with their account on the Manuscript Tracking System (MTS), prior to acceptance. ORCID helps the scientific community achieve unambiguous attribution of all scholarly contributions. You can create and link your ORCID from the home page of the MTS by clicking on 'Modify my Springer Nature account'. For more information, please visit www.springernature.com/orcid.

Thank you for the opportunity to review your work.

Sincerely,
Kyle

Kyle Vogan, PhD
Senior Editor
Nature Genetics
<https://orcid.org/0000-0001-9565-9665>

Referee expertise:

Referee #1: Genetics, complex traits, metabolic disorders

Referee #2: Genetics, complex traits, rare variants

Referee #3: Genetics, complex traits, metabolic disorders

Referee #4: Genetics, complex traits, metabolic disorders

Reviewers' Comments:

Reviewer #1:
Remarks to the Author:

Zhao and colleagues in their paper titled "Population scale whole genome sequencing provides novel insights into cardiometabolic health", explore the burden test yield from 708,956 individuals with whole genome sequencing (WGS) in body mass index (BMI) and Type 2 diabetes (T2D). The authors identified 21 genes (10 in BMI, 12 in T2D, 1 shared; $P < 6.15 \times 10^{-7}$) using burden tests of rare (AAF < 0.1%) variants in 19,457 genes with 3 annotation sets (PTVs, missense [REVEL 0.5 or 0.7]), claim some as novel, and discuss the biological significance of said genes.

Though "selling" the work as illustrating the value of whole genome sequencing, the paper barely scratches the surface of that topic, focusing more on the biology of the diseases. So, while still of broad interest -- as obesity and diabetes are major areas of research -- and the results are mechanistically interesting, the substance of the work is probably less valuable to other readers than one might hope given the size of the data, title, and abstract.

The depth of the work is limited, especially for a whole-genome sequencing (WGS) based study, only looking at protein-coding genes/variants. It makes no attempts to capitalize on the major benefits of WGS over exome sequencing, i.e., non-coding regions, better data for identifying CNVs, and even given the focus of the work on direct comparison of discoveries between WGS and WES the analyses seem basic and insufficient. See comment #1 below.

Finally, as we try to list below, a great deal of detail in the methods of this work are missing in order to truly and thoroughly interpret the analyses and results.

Major questions / concerns

1) The authors note the chi-square statistics were 29% higher for the WGS data than for the WES data as well as the fact that WGS identified more coding variants than WES, and that these unique variants are more likely associated with the traits of interest, but don't provide any explanation or evidence as to why. Simple questions, such as the following, would help exemplify the value of WGS, improve the understanding of the results, and potentially provide valuable insight into gene function in these diseases. All of these analyses would both strengthen the paper's argument and significantly broaden the scope and interest of the work.

a. What number/proportion of genes are captured by WGS but not WES?

b. What number/proportion of exons are captured by WGS but not WES?

c. How many genes are affected by exons lost in WES compared to WGS?

d. Are these the factors leading to the new variants and thus new/stronger associations? Or is it sequence depth or other factors?

e. Can the differences be attributed to different alignment/calling/QC parameters?

2) Two genes claimed as novel (RIF1 and HNF4A), have already been associated at a similar Bonferroni significance threshold via rare variant gene-based tests in the UKB 400k exomes from Karczewski et al., Cell Genomics 2022 (app.genebase.org). Perhaps most notably is RIF1, which was already associated with BMI ($P=3.6e-8$) (<https://app.genebase.org/gene/ENSG00000080345?burdenSet=pLoF&phewasOpts=1&resultLayout=full>). Similarly, HNF4A has a P -value = $1.74e-6$ from SKAT-O in non-insulin diabetes mellitus (<https://app.genebase.org/gene/ENSG00000101076?burdenSet=missense%7CLC&phewasOpts=1&resultLayout=full>). As such, take care in ascribing novelty to the findings.

3) On line 119, the authors write that prior GWAS studies found loci within 500kb of RIF1 and TNK2. It would be good to know if these burden tests are independent of common variants and ideally, if all burden tests were conditionally independent of GWAS loci (I did not see anything in the paper indicating this). Previous studies (e.g., Backman et al., Nature 2022; Hawkes et al., bioRxiv 2023) demonstrated rare variants and gene-based tests can be in partial LD with common variants and no longer be significant after conditioning on GWAS loci. This is a significant caveat of this paper and makes it difficult to judge if these are either real rare variant associations or if they simply reflect linkage to common associations.

4) Burden heritability regression (Weiner, Nadig et al., Nature 2023) is becoming a staple analysis for rare variant studies and is currently missing from this paper. Showing improved rare variant heritability estimates would be a significant additional analysis bolster the authors claim about the importance of WGS. This analysis would also give the people who think deeply about heritability estimates a reason to read the paper.

5) It has been well established that additional genes are missed by burden tests that can be identified by SKAT-like tests. Given this paper's landmark sample size with WGS, it would be interesting to note how well SKAT-O compares to burden tests and whether once again, WGS is improving gene discovery yields beyond burden tests. This analysis may provide additional novel genes that would add a bit of extra weight to the paper. If the authors feel this is not worth the effort and/or outside the scope of the paper, it should be listed as a caveat (which is a section currently missing).

6) The authors should refer to the recent NASEM report on population descriptors when referring to genetic ancestry groups and change their language accordingly when possible. They currently mix ethnic designations (Admixed American/Latino) with geographic designations (European) on lines 540-542. In addition, the NASEM report suggests ancestry descriptors be more specific to their reference data, such as "1kG-like AFR", etc. Furthermore, the methods section currently does not describe how was ancestry defined in AllofUs to note the ancestry sample sizes in the main manuscript and Table S3.

Minor questions / points

1) There seems to be a contradiction regarding whether AllofUs is WGS (based on line 109: "WGS data derived from 219,015 participants in the All of Us studies.") or WES (based on line 538 in the AllofUs methods section: "short-read whole exome sequencing data"). This is most likely a typo, but if not, it seems less than ideal to replicate purported "genome unique" associations with exomes. Please clarify.

2) In addition, especially since the replication dataset was smaller than the discovery and replication efficiency seems rather low, an assessment of statistical power for replication seems appropriate. This may help lend more faith to the non-replicating associations, especially the failed true positives, like GCK.

3) Why are different analysis methods use between the cohorts: BOLT-LMM used for UKB, but STAAR used for AllofUs? How comparable are the two methods and did this influence the replication results?

4) Given that the authors analyzed all samples together irrespective of their ancestry, how did they define the allele frequency used to select variants to be included in the burden tests? Was it the overall allele frequency across all samples,

or the maximum across ancestries, or something else? What about across studies, especially given the dramatically different makeup of AllofUs compared to UKBB?

5) With 3 different gene-based tests performed (PTVs and PTVs+missense at two REVEL levels) on ~20,000 genes, why is the total ~40k per trait and not 60k? If the two missense thresholds are very similar, why do both tests but not adjust for both?

6) Why did the authors use different numbers of PCs (20 PCs for UKB [line 515]; 16 for AoU [line 555])?

7) On line 478-479, the authors mention 3 variant annotations they consider to be PTVs: stop-gained, splice acceptor, and splice donor. Were frameshifts not considered PTVs? Were indels not considered at all?

8) Line 515: age2 (I assume this is age2)? This should have a superscript.

9) Line 516-517: how many WGS batches? Were these included as covariates?

10) Gene names should be italicized in Figures 2-4

11) Error bar descriptions are missing from the legends for Figures 2, 4, and 5.

In sum, while the topic is of value to a large audience, and the biological conclusions noted in the manuscript are interesting, there are many unanswered questions that could both improve the understanding and interpretability of the current work, and substantially broaden interest to a much wider audience.

Reviewer #2:

Remarks to the Author:

Zhao and colleagues in their paper titled "Population scale whole genome sequencing provides novel insights into cardiometabolic health", explore the burden test yield from 708,956 individuals with whole genome sequencing (WGS) in body mass index (BMI) and Type 2 diabetes (T2D). The authors identified 21 genes (10 in BMI, 12 in T2D, 1 shared; $P < 6.15 \times 10^{-7}$) using burden tests of rare (AAF < 0.1%) variants in 19,457 genes with 3 annotation sets (PTVs, missense [REVEL 0.5 or 0.7]), claim some as novel, and discuss the biological significance of said genes.

Though "selling" the work as illustrating the value of whole genome sequencing, the paper barely scratches the surface of that topic, focusing more on the biology of the diseases. So, while still of broad interest -- as obesity and diabetes are major areas of research -- and the results are mechanistically interesting, the substance of the work is probably less valuable to other readers than one might hope given the size of the data, title, and abstract.

The depth of the work is limited, especially for a whole-genome sequencing (WGS) based study, only looking at protein-coding genes/variants. It makes no attempts to capitalize on the major benefits of WGS over exome sequencing, i.e., non-coding regions, better data for identifying CNVs, and even given the focus of the work on direct comparison of discoveries between WGS and WES the analyses seem basic and insufficient. See comment #1 below.

Finally, as we try to list below, a great deal of detail in the methods of this work are missing in order to truly and thoroughly interpret the analyses and results.

Major questions / concerns

1) The authors note the chi-square statistics were 29% higher for the WGS data than for the WES data as well as the fact that WGS identified more coding variants than WES, and that these unique variants are more likely associated with the traits of interest, but don't provide any explanation or evidence as to why. Simple questions, such as the following, would help exemplify the value of WGS, improve the understanding of the results, and potentially provide valuable insight into gene function in these diseases. All of these analyses would both strengthen the paper's argument and significantly broaden the scope and interest of the work.

a. What number/proportion of genes are captured by WGS but not WES?

b. What number/proportion of exons are captured by WGS but not WES?

c. How many genes are affected by exons lost in WES compared to WGS?

d. Are these the factors leading to the new variants and thus new/stronger associations? Or is it sequence depth or other factors?

e. Can the differences be attributed to different alignment/calling/QC parameters?

2) Two genes claimed as novel (RIF1 and HNF4A), have already been associated at a similar Bonferroni significance threshold via rare variant gene-based tests in the UKB 400k exomes from Karczewski et al., Cell Genomics 2022 (app.genebase.org). Perhaps most notably is RIF1, which was already associated with BMI ($P = 3.6 \times 10^{-8}$)

(<https://app.genebase.org/gene/ENSG00000080345?burdenSet=pLoF&phewasOpts=1&resultLayout=full>). Similarly, HNF4A has a P-value = $1.74e-6$ from SKAT-O in non-insulin diabetes mellitus (<https://app.genebase.org/gene/ENSG00000101076?burdenSet=missense%7CLC&phewasOpts=1&resultLayout=full>). As such, take care in ascribing novelty to the findings.

3) On line 119, the authors write that prior GWAS studies found loci within 500kb of RIF1 and TNK2. It would be good to know if these burden tests are independent of common variants and ideally, if all burden tests were conditionally independent of GWAS loci (I did not see anything in the paper indicating this). Previous studies (e.g., Backman et al., Nature 2022; Hawkes et al., bioRxiv 2023) demonstrated rare variants and gene-based tests can be in partial LD with common variants and no longer be significant after conditioning on GWAS loci. This is a significant caveat of this paper and makes it difficult to judge if these are either real rare variant associations or if they simply reflect linkage to common associations.

4) Burden heritability regression (Weiner, Nadig et al., Nature 2023) is becoming a staple analysis for rare variant studies and is currently missing from this paper. Showing improved rare variant heritability estimates would be a significant additional analysis bolster the authors claim about the importance of WGS. This analysis would also give the people who think deeply about heritability estimates a reason to read the paper.

5) It has been well established that additional genes are missed by burden tests that can be identified by SKAT-like tests. Given this paper's landmark sample size with WGS, it would be interesting to note how well SKAT-O compares to burden tests and whether once again, WGS is improving gene discovery yields beyond burden tests. This analysis may provide additional novel genes that would add a bit of extra weight to the paper. If the authors feel this is not worth the effort and/or outside the scope of the paper, it should be listed as a caveat (which is a section currently missing).

6) The authors should refer to the recent NASEM report on population descriptors when referring to genetic ancestry groups and change their language accordingly when possible. They currently mix ethnic designations (Admixed American/Latino) with geographic designations (European) on lines 540-542. In addition, the NASEM report suggests ancestry descriptors be more specific to their reference data, such as "1kG-like AFR", etc. Furthermore, the methods section currently does not describe how was ancestry defined in AllofUs to note the ancestry sample sizes in the main manuscript and Table S3.

Minor questions / points

1) There seems to be a contradiction regarding whether AllofUs is WGS (based on line 109: "WGS data derived from 219,015 participants in the All of Us studies.") or WES (based on line 538 in the AllofUs methods section: "short-read whole exome sequencing data"). This is most likely a typo, but if not, it seems less than ideal to replicate purported "genome unique" associations with exomes. Please clarify.

2) In addition, especially since the replication dataset was smaller than the discovery and replication efficiency seems rather low, an assessment of statistical power for replication seems appropriate. This may help lend more faith to the non-replicating associations, especially the failed true positives, like GCK.

3) Why are different analysis methods use between the cohorts: BOLT-LMM used for UKB, but STAAR used for AllofUs? How comparable are the two methods and did this influence the replication results?

4) Given that the authors analyzed all samples together irrespective of their ancestry, how did they define the allele frequency used to select variants to be included in the burden tests? Was it the overall allele frequency across all samples, or the maximum across ancestries, or something else? What about across studies, especially given the dramatically different makeup of AllofUs compared to UKBB?

5) With 3 different gene-based tests performed (PTVs and PTVs+missense at two REVEL levels) on ~20,000 genes, why is the total ~40k per trait and not 60k? If the two missense thresholds are very similar, why do both tests but not adjust for both?

6) Why did the authors use different numbers of PCs (20 PCs for UKB [line 515]; 16 for AoU [line 555])?

7) On line 478-479, the authors mention 3 variant annotations they consider to be PTVs: stop-gained, splice acceptor, and splice donor. Were frameshifts not considered PTVs? Were indels not considered at all?

8) Line 515: age2 (I assume this is age2)? This should have a superscript.

9) Line 516-517: how many WGS batches? Were these included as covariates?

10) Gene names should be italicized in Figures 2-4

11) Error bar descriptions are missing from the legends for Figures 2, 4, and 5.

In sum, while the topic is of value to a large audience, and the biological conclusions noted in the manuscript are interesting, there are many unanswered questions that could both improve the understanding and interpretability of the current work, and substantially broaden interest to a much wider audience.

Reviewer #3:

Remarks to the Author:

This is an excellent paper that is adding to the small but growing area of using sequenced datasets of 100,000s of people to identify variants present in only a few hundred people but that have very interesting and large effects. There are new genes for BMI and T2D and this combined with the approach is a very important advance in this reviewer's opinion.

I have only minor suggestions:

1. After the sequencing results, the authors bring in GWAS comparisons from the GWAS and eQTL literature, but they bring in both T2D and BMI results even though all the sequencing results are specific (or seem to be) to either BMI or T2D, except UBR3. For example, RIF1 is a BMI gene but only the GWAS signal for T2D is quoted. RMC1 and UBB are linked to T2D, but the BMI GWAS signal is quoted. It might be worth limiting to the same trait – especially as there are 1000s of GWAS signals now so there is a risk of some co-locating by chance? More specifically, on lines 124-127, the authors state that the eQTL colocalises and shows consistent direction with BMI, but it is a T2D variant not a BMI variant? Looking at ST5, there is no BMI signal for RIF1 locus – and this seems more relevant information than the fact that there is a signal for T2D? In the main text, some further clarification could be helpful. Similarly, I would be more interested in IP6K than UBB given the T2D GWAS signal colocation (despite not replicating in All of Us), but this is lost a little if also quoting their BMI GWAS colocation findings.
2. Lines 254-255 - I don't follow what the authors found for TG:HDL - what do they mean by « inconsistent effects » and « which was not regulated » ?
3. Figure 2 might be worth some additional annotation? Whilst most of the genes are real - either from convincing replication in All of Us or known biology - some are far less certain – perhaps emboldened the ones that replicate in AOU?
4. Figure 5. For better visualisation, it might be worth scaling the x axis so that the null effects (0 for BMI effect; 1 for T2D) are aligned).

Reviewer #4:

Remarks to the Author:

Review

Zhao et al., "Population-scale whole genome sequencing provides novel insights into cardiometabolic health"

Summary of key results: The authors use UKBB WGS data 490K from coding regions to conduct rare-variant gene-burden tests focused on protein-truncating variants (PTV) or missense variants for BMI and T2D. They compare their findings with their previous work in WES (in UKB 450K). By this, they find novel genes with rare-variant-burden for these traits that the authors say was not only due to the increased sample size, but also due to the technical platform (WGS rather than WES). The authors provide independent replication data from All-of-Us study and evidence for replication at $P < 0.05$ for five novel genes with rare-variant-burden for BMI and/or T2D: from overall 21 genes identified with rare-variant-burden, there were novel gene findings (WGS versus WES) for 3 genes (for BMI; UBR2, RIF2, and TNK2) and 7 genes (for T2D), of which 2 genes (BMI; UBR2, RIF2) and 3 genes (T2D; IRS2, UBR2, HNF4A) were replicated (one overlapping; where is the fifth?).

The authors conduct various follow-up analyses using other metabolic phenotypes available in UKB to validate phenotypes reported in mouse by human associations. This is a very interesting approach, as this can help understand mediating effects. For IRS2, they find an independent association with kidney function (eGFR) and chronic kidney disease (CKD, probably defined as $eGFR < 60$ units?). There are some questions (see below) that can probably be easily clarified; a description of the analyzed UKB and All-of-Us dataset regarding age, sex, T2D, CKD, BMI, etc. is needed as is a clear definition of T2D and CKD for both studies.

Methodologically, the authors state that WGS results for coding regions can be advantageous over WES, which would benefit from a main figure to support this conclusion for all readers (not only Supplementary Table).

The manuscript is overall well structured and well written. The results are clearly provided in Supplementary Tables and key findings in Figures.

Originality and Significance: The authors say that this is the first genome-wide multi-ancestry gene burden test using WGS from > 700,000 individuals. There are several novel interesting findings for BMI and T2D and consequences on other metabolic parameters using human association results to validate phenotypes observed in mouse. The results are interesting to a wider audience, when the WGS-based rare-variant-burden results are provided for T2D and BMI genome-wide (data availability).

Data and methodology: The data is UK Biobank and well-known for high quality, as is All-of-Us. The methods are state-of-the-art, and the evidence provided in Figures and Supplementary Tables is convincing and comprehensive.

Statistics and treatment of uncertainties: The appropriate statistics are generated and provided; some overview rather laundry lists of statistics in results text (first part of results, see details below) might be helpful to the reader. There is no critical appraisal of potential uncertainties in the definition of T2D or CKD and how this might affect results in UKB, where individuals are rather healthy.

Conclusions: A replication of the association of IRS-2 with CKD independent of T2D and all downstream mediator analyses in All-of-Us, which has probably more and clinically better-defined T2D and CKD, could support the findings and the strong statements made in the discussion. UKB individuals are rather healthy and CKD and T2D are probably (not clear from methods) both made on blood biomarkers rather than a clinical diagnosis. This is OK for association analyses, but the conclusions refer to clinically diagnosed patients, which might be a bit of a stretch as it stands now.

Clarity and context: The title does not appear to catch the key approach and result of the paper. The manuscript is about gene-burden tests based on WGS addressing coding regions that helps prioritize genes for metabolic health. Is this cardio-metabolic or just metabolic? There is nothing about CAD or CVD. Is “novel insights” and the rest of the title precise enough to distinguish this paper from others and to catch the key approach and message of the manuscript?

Specific suggestions for improvements: Abstract and introduction are fine.

Results:

(1) Please provide participant characteristics in a Table for the analyzed UKBB individuals and All-of-Us including stats on the distribution of age, sex, ancestry, BMI, %T2D, how T2D was defined in UKBB (state that All-of-Us use medical records for T2D), distribution of eGFR, %CKD (and how CKD was defined). Is All-of-Us based on hospital records (or from general practitioner)?

(2) Lines 100-103: “we observed stronger associations using WGS than ... using WES, ... 29% increase in mean chi-square”: this is methodologically interesting. Please clarify the sample size (#individuals) for this comparison in the text (here it appears to be different sample size). A clearer presentation of this comparison (not only by lists in Supplementary Table 2) might be scatter plots of effect sizes (beta or OR, respectively), their standard (SE) standard errors, and #variants for each identified gene burden. A sole comparison of the change in Chi-Square statistics does not help dissect power by smaller SEs versus power due to larger effect sizes or power due to larger number of variants in the burden test.

(3) This relates to lines 166-183: the authors appear to repeat this comparison now in the same sample size: please state this more clearly adding the sample size in the main text. Line 180: “In contrast, some variants were identified by WES-only”: so, on average, WGS included more variants than WES, but WGS also missed some variants? A graphical presentation of these results (details are already in Supplementary Table 2) might be helpful, given that the conclusion (“... quantify the enhanced coverage of coding variants provided by WGS over WES in UK Biobank”) is a main message.

(4) Lines 92-160: There are several “laundry lists” on identified genes without and with replication, confirmed and known (a lot of stats in parentheses) in the results text (lines 92-160). Detailed results are in Supplementary Table and an overview is in Figure 2: Could beta-estimates for BMI and OR estimates (as numbers) be added to the Figure 2, together with carrier n and case prevalence? And replicated genes marked in bold? Provide a 2nd panel to show previous WES results on these? Then a large part of these laundry lists in the text can be avoided and a much better understanding of the findings (and their novelty) can be appreciated.

(5) Lines 120-128: It is very interesting that the authors link rare-variant-burden genes with GWAS loci. Can the authors rule out a shadow effect where the common GWAS-variant tags a haplotype with many of the rare alleles in the burden test? Then the rare-variant-burden can be significant even if the rare variants do not exert a causal effect? Specifically, what is the rare-variant-burden when adjusting for the GWAS lead variant (effect estimate and P-value) compared to unadjusted? Or is it the other way round: does the rare-variant-burden explain the GWAS signal? (i.e. does the GWAS signal disappear when adjusted for the rare-variant-burden).

(6) Supplementary Table 5, cited from line 128: For the region around each rare-variant-burden identified gene, could the variant with the smallest P-value and its P-value from GWAS be added?

(7) Would it be helpful to separate out some of the first chapter (no subtitle) into the primary analyses (identifying novel gene burden) and the more methodological follow-up (LOO sensitivity analyses and the exploratory analyses of comparing WGS versus WES in sample size)?

A phenotypic association scan of identified genes ... novel role for IRS2

(8) Is the IRS2 association stated as beta = -10.42 for eGFR and OR = 4.0 for CKD adjusted for T2D? Could you please provide both the T2D-unadjusted and T2D-adjusted effects estimates and P-values in the main text and Figure 4? (One caveat here is that this reviewer could not see Supplementary Figure 3; nor any Supplementary Figure).

(9) It is an interesting finding to link IRS2 with kidney function independent of T2D. “this association does not simply reflect the consequences of T2D-mediated chronic hyperglycaemia”: What exactly is the definition of T2D used here for UKB?

(10) To show these results for 3 different equations for eGFR is to be appreciated, but: How was CKD was defined?

(eGFR_{crea}<60?) Please add n of CKD cases and controls (to Figure 4 legend). UK Biobank is rather healthy with a small proportion of CKD among individuals. Can this be replicated in All-of-Us with probably more and clinically better-defined CKD patients?

Evidence of ... IRS1/IRS2 mediated signalling

(11) Again, it is very interesting to try validated mouse phenotypes by human association data.

a. Figure 5 legends state "IRS-1 reduces body size with modest effect on blood glucose", but "height" and "fat free mass" and "T2D" are shown: there are various measures of "body size" beyond height (just call it "height"?), is fat free mass a measure of "body size" or a mixture of muscle and bone? Where is the "modest effect on blood glucose"? does this refer to the association with T2D? but T2D status is not necessarily reflecting blood glucose, if treated. Better stick to the concrete variable name, at least in results text and Figure legends.

b. Also "IRS-2 causes severe hyperglycaemia" without affecting body size: the term "cause" is difficult in epidemiological studies (please avoid); the association is with T2D, which typically includes antidiabetically treated (and not hyperglycemic at the study center assessment). Is the association of IRS2 with T2D persistent when adjusted for height and fat-free-mass? Is there still no association of IRS1 on T2D when adjusted for height and fat-free-mass?

UBR2 and UBR3

(12) The OR for T2D for UBR3 was 3.4 in UKB without adjustment for BMI ($P=6 \times 10^{-9}$) and 2.5 ($P=2.7 \times 10^{-4}$) with adjustment: could you re-state the unadjusted results? In All-of-Us, the OR was 2.7 without adjustment: what is it in All-of-Us with adjustment for BMI? Is the definition of T2D in All-of-Us different from the definition in UKB?

(13) There are many interesting results but a condensed view in a Figure on these results on UBR2 and UBR3 would be helpful (similar to the Figure for IRS1 and IRS2).

Methods:

(14) There is a generic explanation of the All-of-Us T2D definition in the methods (page 22, line 546: "concept ID 201826..."), but this is the specification of a code, but not what this means in terms of: HbA1c? anti-diabetic medication? Hospital or GP-based medical record of T2D diagnosis? What does "initial entry" mean (line 548): is this not BMI from height and weight measured at study center baseline and the concordant T2D status at the same time? If BMI and T2D records are not at the same timepoint, how was this accounted for in the analyses?

(15) What IS the definition of T2D in UKB: HbA1c? anti-diabetic medication? how was CKD defined?

Discussion:

(1) page 12, line 328: what do you mean with "algorithmically defined CKD"?

(2) Link of IRS-2 to SIRD: was IRS-2 associated with parameters of insulin resistance? (lines 345-352). This reviewer might have missed this; maybe this can be made a bit clearer in the results or discussed why no association with insulin resistance was found.

Version 1:

Decision Letter:

13th March 2025

Dear John,

Your revised Article "Population-scale gene-based analysis of whole genome sequencing provides novel insights into metabolic health" has been seen by the original referees. You will see from their comments below that, while Reviewers #1-#3 are satisfied with the revision and have no further requests, Reviewer #4 has raised a few ongoing concerns. We remain interested in the possibility of publishing your study in Nature Genetics, but we would like to consider your response to these ongoing concerns in the form of a further revision before we make a final decision on publication.

In particular, we ask that you further clarify how outcomes were defined in each study sample, discuss how misclassifications could impact interpretation of the analyses, and perform additional analyses to adjust for shadow effects of common variants as requested by Reviewer #4.

We therefore invite you to revise your manuscript taking into account all reviewer and editor comments. Please highlight all changes in the manuscript text file. At this stage, we will need you to upload a copy of the manuscript in MS Word .docx or similar editable format.

*2) If you have not done so already, please begin to revise your manuscript so that it conforms to our Article format instructions, available

http://www.nature.com/ng/authors/article_types/index.html>here.

*3) Include a revised version of any required Reporting Summary: <https://www.nature.com/documents/nr-reporting-summary.pdf>

Please be aware of our <https://www.nature.com/nature-research/editorial-policies/image-integrity>>guidelines on digital image standards.

EXTENDED DATA FIGURES

Link Redacted

We hope to receive your revised manuscript within 4-8 weeks. If you cannot send it within this time, please let us know.

Nature Genetics is committed to improving transparency in authorship. As part of our efforts in this direction, we are now requesting that all authors identified as 'corresponding author' on published papers create and link their Open Researcher and Contributor Identifier (ORCID) with their account on the Manuscript Tracking System (MTS), prior to acceptance. ORCID helps the scientific community achieve unambiguous attribution of all scholarly contributions. You can create and link your ORCID from the home page of the MTS by clicking on 'Modify my Springer Nature account'. For more information, please visit <http://www.springernature.com/orcid>>www.springernature.com/orcid.

Sincerely,
Kyle

Kyle Vogan, PhD
Senior Editor
Nature Genetics
<https://orcid.org/0000-0001-9565-9665>

Referee expertise:

Referee #1: Genetics, complex traits, metabolic disorders

Referee #2: Genetics, complex traits, rare variants

Referee #3: Genetics, complex traits, metabolic disorders

Referee #4: Genetics, complex traits, metabolic disorders

Reviewers' Comments:

Reviewer #1 (Remarks to the Author):

The authors met an extensive and comprehensive set of questions and comments across the reviews. They have thoughtfully and thoroughly addressed these issues to better and more accurately address each of their topics of interest. I have little doubt the work will be of interest both mechanistically/biologically and in more directly addressing the comparative value of WES/WGS analyses. I commend the team for their detailed approach to managing and addressing the extensive reviews.

I have no outstanding issues or concerns with this work.

Reviewer #2 (Remarks to the Author):

Co-reviewed with Reviewer #1.

Reviewer #3 (Remarks to the Author):

The authors have done a great job addressing the reviewers' questions. I think the more in-depth analysis of the reasons behind the higher yield of signals from WGS vs WES is an important addition. The 4.6% increase from increased sample size and 21% from more variants seems consistent with other reports that WGS results in about 10% increase in information, given that most of the increase in sample size comes from more diverse samples...so the two are related. The change in tone in the title also makes sense given the main message is about new genes not so much WGS vs WES or other approaches.

Reviewer #4 (Remarks to the Author):

Zhao et al. "Population-scale gene-based analysis of whole exome sequencing provides novel insights into metabolic health"

The authors have answered to all comments. However, two concerns remain.

One concern is the still not fully clear description of the outcome variables:

Regarding the definition of CKD in UKBB, the authors answer to a first comment, that hospital episode statistics were used. However, CKD does not typically lead to hospitalization. Hospital records are insufficient for this. To a later comment and methods text, it seems that they used UKBB records from general practitioners (GPs), which makes sense. However, the methods' text on how their main outcome variables were generated does not allow for an understanding of what these outcome definitions means and how well ascertained these can be deemed. The same applies to the diagnosis of T2D. For T2D, this seems to be similar for All-of-Us but was not fully clear. The CKD finding could have been replicated in All-of-Us, as the data is available in that study, but this was not done? The term "algorithmically derived CKD" used at very instances in the text is uninformative, as indicated previously by this reviewer, but is still used.

To the question about the potential impact of misclassification in T2D or CKD, the authors answer that "uncertainty in case identification is expected to weaken the strength of any true effect rather than causing inflation or bias": this is true when the misclassification is in the outcome, but this is not true when the misclassification is in the covariable used to rule-out a mediating effect. A critical appraisal of the limitations in their case/control definitions and how this may affect their results and conclusions is still lacking.

The second concern refers to the problem that rare-variant-burden can arise from a shadow effect by a common variant associated with the trait that resides in or near the gene in question. When rare alleles sit on a haplotype of a common variant allele, and that common variant is associated with the outcome, the rare-variant-burden can be significant, but this does not support the gene as causal. It is mandatory to exclude a shadow effect and only genes where the rare-variant-burden remains significant after adjustment can be deemed to support the gene as causal. The authors deemed their analysis to adjust for common variant effects as "sensitivity analyses", but it is more than that: the unadjusted analyses can be just a first indicator; the adjusted analyses are the valid analysis.

Even more of concern is the authors approach of how to adjust the rare-variant-burden for a shadow effect. They use a polygenic risk score (made across some common variants associated with the outcome). However, some polygenic score for the outcome is not sufficient to use here: the region (with their WGS-data) needs to be tested for single-variant-association and the significantly associated variants from this need to be used for the adjustment to rule out a shadow effect of the rare-variant-burden. The polygenic score might or might not include any variant of that region; and very likely does not contain all the variants that are associated with the outcome in single-variant testing. Thus, the rare-variant burden of the 21 claimed genes and particularly the three genes newly identified for BMI and the seven genes newly identified for T2D needs to be

substantiated by adjusting for variants associated with the respective outcome in single-variant testing. As the results stand, it is not clear whether their claimed genes are truly supported as causal genes by rare-variant-burden or mere shadow effects.

Version 2:

Decision Letter:

Our ref: NG-A65633R1

11th June 2025

Dear John,

Thank you for submitting your revised manuscript "Population-scale gene-based analysis of whole genome sequencing provides novel insights into metabolic health" (NG-A65633R1). In light of the revisions made in response to Reviewer #4, we will be happy in principle to publish your study in Nature Genetics as an Article pending final revisions to comply with our editorial and formatting guidelines.

We are now performing detailed checks on your paper, and we will send you a checklist detailing our editorial and formatting requirements soon. Please do not upload the final materials or make any revisions until you receive this additional information from us.

Thank you again for your interest in Nature Genetics. Please do not hesitate to contact me if you have any questions.

Sincerely,
Kyle

Kyle Vogan, PhD
Senior Editor
Nature Genetics
<https://orcid.org/0000-0001-9565-9665>

Thank you for the comments which have prompted several improvements to the paper. All of the comments have been addressed, as detailed below.

Reviewer #1:

Remarks to the Author:

Zhao and colleagues in their paper titled "Population scale whole genome sequencing provides novel insights into cardiometabolic health", explore the burden test yield from 708,956 individuals with whole genome sequencing (WGS) in body mass index (BMI) and Type 2 diabetes (T2D). The authors identified 21 genes (10 in BMI, 12 in T2D, 1 shared; $P < 6.15 \times 10^{-7}$) using burden tests of rare (AAF < 0.1%) variants in 19,457 genes with 3 annotation sets (PTVs, missense [REVEL 0.5 or 0.7]), claim some as novel, and discuss the biological significance of said genes

Though "selling" the work as illustrating the value of whole genome sequencing, the paper barely scratches the surface of that topic, focusing more on the biology of the diseases. So, while still of broad interest -- as obesity and diabetes are major areas of research -- and the results are mechanistically interesting, the substance of the work is probably less valuable to other readers than one might hope given the size of the data, title, and abstract.

The depth of the work is limited, especially for a whole-genome sequencing (WGS) based study, only looking at protein-coding genes/variants. It makes no attempts to capitalize on the major benefits of WGS over exome sequencing, i.e., non-coding regions, better data for identifying CNVs, and even given the focus of the work on direct comparison of discoveries between WGS and WES the analyses seem basic and insufficient. See comment #1 below.

Response: We thank the reviewer for their comments that our results are 'of broad interest...and mechanistically interesting'. In this regard, we achieved our goal to use the WGS data to provide novel insights into the biology of cardiometabolic health. While we agree that there are also likely many interesting non-coding associations, annotating the causal genes/mechanisms remains a substantial challenge for such approaches. We therefore focussed on protein-coding variant associations and now describe in more detail how the WGS data in UK Biobank adds to previous WES data.

Finally, as we try to list below, a great deal of detail in the methods of this work are missing in order to truly and thoroughly interpret the analyses and results.

Major questions / concerns

1) The authors note the chi-square statistics were 29% higher for the WGS data than for the WES data as well as the fact that WGS identified more coding variants than WES, and that these unique variants are more likely associated with the traits of interest, but don't provide any explanation or evidence as to why. Simple questions, such as the following, would help exemplify the value of WGS, improve the

understanding of the results, and potentially provide valuable insight into gene function in these diseases. All of these analyses would both strengthen the paper's argument and significantly broaden the scope and interest of the work.

- a. What number/proportion of genes are captured by WGS but not WES
- b. What number/proportion of exons are captured by WGS but not WES?
- c. How many genes are affected by exons lost in WES compared to WGS?
- d. Are these the factors leading to the new variants and thus new/stronger associations? Or is it sequence depth or other factors?
- e. Can the differences be attributed to different alignment/calling/QC parameters?

Response: Much of a-e has been described in the "flagship" UKBB WGS paper, which highlighted a couple of examples (*PKHD1* & *LPA*) where they state: "the number of samples with $\geq 10X$ coverage drops in the WES compared to WGS DRAGEN dataset at specific coding region (CDS) sites/exons (Fig. S10). For example, for *LPA* only $\sim 39\%$ of samples achieve $\geq 10X$ coverage across exons 10-15 (bp: 160,612,932 -160,628,366)" see - <https://www.medrxiv.org/content/10.1101/2023.12.06.23299426v1.full-text> .

We illustrate in the Figure below the improved coverage of *IRS2* by WGS vs WES:

Figure: Genomic location of PTVs in *IRS2* and number of samples with at least 10X coverage across CDS sites in the WES and WGS.

We had calculated that our WGS analysis produced overall 29% higher mean chi-square values vs. WES across our highlighted genes. We now add that this increase seems due to the larger all ancestry sample (4.6% higher mean chi-square) and the higher number of variants found by WGS (21% higher mean chi-square). Furthermore, as also suggested by Reviewer 4, Results comment #2, we have added Supplementary Figures to show there are no differences in effect estimates

on WGS and WES (see below, and added as **Supplementary Figures 1-2, Supplementary Table 5**).

These data are amalgamated in a new results section (Page 7) subtitled: **“Increased power using an all ancestry WGS approach”**.

2) Two genes claimed as novel (RIF1 and HNF4A), have already been associated at a similar Bonferroni significance threshold via rare variant gene-based tests in the UKB 400k exomes from Karczewski et al., Cell Genomics 2022 (app.genebass.org). Perhaps most notably is RIF1, which was already associated with BMI ($P=3.6e-8$) (<https://app.genebass.org/gene/ENSG00000080345?burdenSet=pLoF&phewasOpts=1&resultLayout=full>). Similarly, HNF4A has a P-value = $1.74e-6$ from SKAT-O in non-insulin diabetes mellitus (<https://app.genebass.org/gene/ENSG00000101076?burdenSet=missense%7CLC&phewasOpts=1&resultLayout=full>). As such, take care in ascribing novelty to the findings.

Response: We strongly consider that such automated PheWAS outputs may provide useful exploratory results, but they produce huge catalogues of results without any validation, biological interpretation or context. We already have millions of genetic associations which we do not understand in any meaningful way. Our approach – combining domain specific knowledge with statistical genetics expertise in a narrow range of disease areas - is focussed on identifying robust genetic associations that provide insight into the biology of these diseases and function of the highlighted genes. Our results are the first dedicated ExWAS of T2D and BMI to report these associations (note these results were not discussed or highlighted in the cited paper).

3) On line 119, the authors write that prior GWAS studies found loci within 500kb of RIF1 and TNK2. It would be good to know if these burden tests are independent of common variants and ideally, if all burden tests were conditionally independent of GWAS loci (I did not see anything in the paper indicating this). Previous studies (e.g., Backman et al., Nature 2022; Hawkes et al., bioRxiv 2023) demonstrated rare variants and gene-based tests can be in partial LD with common variants and no longer be significant after conditioning on GWAS loci. This is a significant caveat of this paper and makes it difficult to judge if these are either real rare variant associations or if they simply reflect linkage to common associations.

Response: As suggested, we have added linear mixed models that include polygenic risk scores for T2D and BMI as covariates. Of our 5 confirmed novel gene-disease associations, four were modestly attenuated (but retain exome-wide significance) and one was modestly strengthened (HNF4A-T2D). These results are added to Supplementary table 3.

“We also tested if any of our rare variant discoveries were ‘tagged’ by common variant associations. We generated polygenic risk scores for each trait and included

these as co-variates. Of our 5 confirmed novel gene-disease associations, four were modestly attenuated (but retain exome-wide significance) and one was modestly strengthened (HNF4A-T2D). These results indicate that the identified rare variant effects on Type 2 Diabetes and BMI are independent of common variants.” (Page 6, Line 138-143)

4) Burden heritability regression (Weiner, Nadig et al., Nature 2023) is becoming a staple analysis for rare variant studies and is currently missing from this paper. Showing improved rare variant heritability estimates would be a significant additional analysis bolster the authors claim about the importance of WGS. This analysis would also give the people who think deeply about heritability estimates a reason to read the paper.

Response: While this suggestion is interesting, we are unsure how to demonstrate that our heritability estimate from WGS data is ‘better’ than previous approaches in the same population. As stated above, our interest is in identifying novel genetic risk factors and providing new insights into the pathophysiology of obesity and type 2 diabetes. Generating heritability estimations and contrasting with other ways of assaying genetic variation is outside the scope of this work and better placed in another more methodological paper.

5) It has been well established that additional genes are missed by burden tests that can be identified by SKAT-like tests. Given this paper’s landmark sample size with WGS, it would be interesting to note how well SKAT-O compares to burden tests and whether once again, WGS is improving gene discovery yields beyond burden tests. This analysis may provide additional novel genes that would add a bit of extra weight to the paper. If the authors feel this is not worth the effort and/or outside the scope of the paper, it should be listed as a caveat (which is a section currently missing).

Response: We avoided SKAT/SKAT-O and related tests as the results are not always easily interpretable and to minimise multiple testing burden. As suggested, we performed SKAT-O based tests implemented in REGENIE. Unfortunately, all-ancestry analyses using REGENIE produce massively inflated test statistics (as evidenced by QQ plots for synonymous masks, see below). We therefore have not pursued this approach further.

BMI - all ancestries combined

Syn Add

Syn SKAT-O

BMI - EU only

Syn Add

Syn SKAT-O

6) The authors should refer to the recent NASEM report on population descriptors when referring to genetic ancestry groups and change their language accordingly when possible. They currently mix ethnic designations (Admixed American/Latino) with geographic designations (European) on lines 540-542. In addition, the NASEM report suggests ancestry descriptors be more specific to their reference data, such as "1kG-like AFR", etc. Furthermore, the methods section currently does not describe how was ancestry defined in AllofUs to note the ancestry sample sizes in the main manuscript and Table S3.

Response: The labels we used are those provided by the All of Us team who centrally performed ancestry assignment. We have added details on their methodology in the methods section and included cohort composition by ancestry in the Supplementary tables. We have now added the corresponding 1KG/HGDP population labels to the relevant supplementary table (**Supplementary Table 17**).

Minor questions / points

1) There seems to be a contradiction regarding whether AllofUs is WGS (based on line 109: "WGS data derived from 219,015 participants in the All of Us studies.") or WES (based on line 538 in the AllofUs methods section: "short-read whole exome sequencing data"). This is most likely a typo, but if not, it seems less than ideal to replicate purported "genome unique" associations with exomes. Please clarify.

Response: Apologies for the typo. All of Us did indeed provide short read whole genome sequencing data. We have corrected this error.

2) In addition, especially since the replication dataset was smaller than the discovery and replication efficiency seems rather low, an assessment of statistical power for replication seems appropriate. This may help lend more faith to the non-replicating associations, especially the failed true positives, like GCK.

Response: As suggested, we have calculated risks of Type 2 errors for the replication sample. After correction for winner's curse, the 3 non-replicating masks for BMI appear to be due to low power (high risk of Type 2 error) and 4 of the 7 non-replicating T2D masks have a type 2 error rate that exceeds 15%.

However, power for replication is not only based on overall sample size. Our replication analysis used identical criteria to define variants masks in All of Us as in UKBB, but the exact included variants could differ depending on their presence and MAF in each sample, e.g. valid functional variants in UKBB might be excluded from replication if not present or $MAF > 0.1\%$ in All of Us. (e.g. failure to replicate PCSK1 and GCK with BMI and T2D which are very likely true positives in UKBB).

We have added these results to supplementary tables 8 and 9 and a section describing these results as below:

*“To understand if failed replication was related to limited statistical power, we conducted power calculations after correction for winner's curse (**see methods**). For BMI, the risk of type 2 error exceeded 30% for all three of the non-replicating masks. For Type 2 Diabetes, 4 of the non-replicating masks had a type 2 error rate exceeding 15%; the other 3 non-replicating masks had adequate power (GCK Missense, REVEL > 0.5, TNRC6B and NAA15 PTVs) (**Supplementary Table 8 & 9**).” (Page 8, Line 193-200)*

3) Why are different analysis methods use between the cohorts: BOLT-LMM used for UKB, but STAAR used for AllofUs? How comparable are the two methods and did this influence the replication results?

Response: We used independently established computational pipelines – UKBB data was analysed by researchers at the University of Cambridge in collaboration with GSK, and All of Us data analysed by researchers at the NIH. We view this as a strength rather than a limitation, but appreciate it complicates the interpretation of non-replicating results. In previous works, we observed good agreement between STAAR and BOLT-LMM [PMID: 36530175, PMID: 34234147, PMID: 34875679, PMID: 38867047] but we are unable to conduct BOLT-LMM in All of Us as establishment of these computational pipelines is laborious and time-consuming.

4) Given that the authors analyzed all samples together irrespective of their ancestry, how did they define the allele frequency used to select variants to be included in the burden tests? Was it the overall allele frequency across all samples, or the maximum

across ancestries, or something else? What about across studies, especially given the dramatically different makeup of All of Us compared to UKBB?

Response: MAF cut-offs were calculated within each studied sample. This has been clarified in the methods.

“Gene-based burden tests were applied to variants with MAF less than 0.001 which met pre-specified bioinformatic criteria and were in selected genes (e.g. those significant in UKBB discovery). Note that due to different population composition, variant MAF will differ between All of Us and UKBB.” (Page 26-27, Line 629-632)

5) With 3 different gene-based tests performed (PTVs and PTVs+missense at two REVEL levels) on ~20,000 genes, why is the total ~40k per trait and not 60k? If the two missense thresholds are very similar, why do both tests but not adjust for both?

Response: We pre-specified a MAC cut-off of 30 for a mask to be included in our analysis, thus the total number of qualifying tests is substantially less than the theoretical total. Our multiple test correction stringently considered all three masks (HC PTV as per LOFTEE, REVEL > 0.5 and REVEL > 0.7) as separate tests.

6) Why did the authors use different numbers of PCs (20 PCs for UKB [line 515]; 16 for AoU [line 555])?

Response: We included all the 16 pre-computed PCs provided centrally in AoU.

7) On line 478-479, the authors mention 3 variant annotations they consider to be PTVs: stop-gained, splice acceptor, and splice donor. Were frameshifts not considered PTVs? Were indels not considered at all?

Response: Apologies for the typo, frameshifts were indeed considered as PTVs. We have corrected this text. indels per se were not considered – except for those that resulted in frameshifts or other qualifying changes.

8) Line 515: age² (I assume this is age²)? This should have a superscript.

Response: Yes. We have corrected this.

9) Line 516-517: how many WGS batches? Were these included as covariates?

Response: There are three batches for UKBB WGS release (Vanguard project, Sanger: 49,932, Sanger: 193,075, deCode: 247,504). We included them as covariates.

*“The covariates included in our analysis are age, age², sex, age*sex, the first 20 principal components as calculated from all WGS samples, and the 3 WGS-released batch (Vanguard project, Sanger: 49,932, Sanger: 193,075, deCode: 247,504).” (Page 25, Line 570-572)*

10) Gene names should be italicized in Figures 2-4

Response: Thanks for this comment. We will work with the editorial team to integrate these amendments if accepted.

11) Error bar descriptions are missing from the legends for Figures 2, 4, and 5.

Response: We have added these to the Figure legends.

In sum, while the topic is of value to a large audience, and the biological conclusions noted in the manuscript are interesting, there are many unanswered questions that could both improve the understanding and interpretability of the current work, and substantially broaden interest to a much wider audience.

Response: We thank the reviewer for their helpful comments which have improved our manuscript.

Reviewer #3:

Remarks to the Author:

This is an excellent paper that is adding to the small but growing area of using sequenced datasets of 100,000s of people to identify variants present in only a few hundred people but that have very interesting and large effects. There are new genes for BMI and T2D and this combined with the approach is a very important advance in this reviewer's opinion.

I have only minor suggestions:

1. After the sequencing results, the authors bring in GWAS comparisons from the GWAS and eQTL literature, but they bring in both T2D and BMI results even though all the sequencing results are specific (or seem to be) to either BMI or T2D, except UBR3. For example, RIF1 is a BMI gene but only the GWAS signal for T2D is quoted. RMC1 and UBB are linked to T2D, but the BMI GWAS signal is quoted. It might be worth limiting to the same trait – especially as there are 1000s of GWAS signals now so there is a risk of some co-locating by chance? More specifically, on lines 124-127, the authors state that the eQTL colocalises and shows consistent direction with BMI, but it is a T2D variant not a BMI variant? Looking at ST5, there is no BMI signal for RIF1 locus – and this seems more relevant information than the fact that there is a signal for T2D? In the main text, some further clarification could be helpful. Similarly, I would be more interested in IP6K than UBB given the T2D GWAS signal colocation (despite not replicating in All of Us), but this is lost a little if also quoting their BMI GWAS colocation findings.

Response: As suggested, we have revised the text and Supplementary Tables to limit GWAS look ups only for the relevant trait.

2. Lines 254-255 - I don't follow what the authors found for TG:HDL - what do they mean by « inconsistent effects » and « which was not regulated » ?

Response: We apologise for the lack of clarity. We analysed two markers of insulin resistance, TG:HDL and SHBG. The former is increased in insulin resistance and the latter is decreased. We observed that TG:HDL was no different in carriers of UBR3 PTVs whereas SHBG was nominally decreased. We have rephrased this text for clarity:

“We found no evidence for an effect of PTVs in UBR3 on body fat distribution as assessed by WHRadjBMI and inconsistent effects on the surrogate markers of insulin resistance - TG:HDL was not regulated but SHBG was nominally decreased.” (Page 11, Line 296-299)

3. Figure 2 might be worth some additional annotation? Whilst most of the genes are real - either from convincing replication in All of Us or known biology - some are far less certain – perhaps emboldened the ones that replicate in AOU?

Response: As suggested, we now indicate the replicated genes in bold text.

4. Figure 5. For better visualisation, it might be worth scaling the x axis so that the null effects (0 for BMI effect; 1 for T2D) are aligned).

Response: We tried this but because the directions of effect are directionally opposing (lower height and fat free mass; higher risk of T2D) this would lead to a lot of empty space and reduced resolution.

Reviewer #4:

Remarks to the Author:

Zhao et al., "Population-scale whole genome sequencing provides novel insights into cardiometabolic health"

Summary of key results: The authors use UKBB WGS data 490K from coding regions to conduct rare-variant gene-burden tests focused on protein-truncating variants (PTV) or missense variants for BMI and T2D. They compare their findings with their previous work in WES (in UKB 450K). By this, they find novel genes with rare-variant-burden for these traits that the authors say was not only due to the increased sample size, but also due to the technical platform (WGS rather than WES). The authors provide independent replication data from All-of-Us study and evidence for replication at $P < 0.05$ for five novel genes with rare-variant-burden for BMI and/or T2D: from overall 21 genes identified with rare-variant-burden, there were novel gene findings (WGS versus WES) for 3 genes (for BMI; UBR2, RIF2, and TNK2) and 7 genes (for T2D), of which 2 genes (BMI; UBR2, RIF2) and 3 genes (T2D; IRS2, UBR2, HNF4A) were replicated (one overlapping; where is the fifth?).

Response: The novel replicated associations were for BMI: *UBR3* and *RIF1*; and for T2D: *IRS2*, *UBR3* and *HNF4A*. Thus, there were 5 novel gene-disease associations across four different genes.

The authors conduct various follow-up analyses using other metabolic phenotypes available in UKB to validate phenotypes reported in mouse by human associations. This is a very interesting approach, as this can help understand mediating effects. For IRS2, they find an independent association with kidney function (eGFR) and chronic kidney disease (CKD, probably defined as $eGFR < 60$ units?). There are some questions (see below) that can probably be easily clarified; a description of the analyzed UKB and All-of-Us dataset regarding age, sex, T2D, CKD, BMI, etc. is needed as is a clear definition of T2D and CKD for both studies.

Response: Chronic Kidney disease in UKBB was defined using the linkage to hospital episode statistics and death certificates, not biochemical criteria. Participants with recorded ICD10, Read and OPSC4 codes consistent with a diagnosis of chronic kidney disease were designated cases, and those without any qualifying conditions were designated controls. A similar approach was used for Type 2 Diabetes. The specific criteria have been appended (as columns N & O) to **Supplementary table 10**.

We have added details on case/control definitions to the methods in the section **"Phenotype preparation in UKBB"**

Methodologically, the authors state that WGS results for coding regions can be advantageous over WES, which would benefit from a main figure to support this conclusion for all readers (not only Supplementary Table).

Response: We have added two supplementary figures and a dedicated section in the text in response to this comment and other reviewer comments.

We had calculated that our WGS analysis produced overall 29% higher mean chi-square values vs. WES across our highlighted genes. We now add that this increase seems due to the larger all ancestry sample (4.6% higher mean chi-square) and the higher number of variants found by WGS (21% higher mean chi-square).

Furthermore, as also suggested by Reviewer 4, Results comment #2, we have added Supplementary Figures to show there are no differences in effect estimates on WGS and WES (see below, and added as **Supplementary Figures 1-2, Supplementary Table 5**).

These data are amalgamated in a new results section (Page 7) subtitled: **“Increased power using an all ancestry WGS approach”**.

The manuscript is overall well structured and well written. The results are clearly provided in Supplementary Tables and key findings in Figures.

Originality and Significance: The authors say that this is the first genome-wide multi-ancestry gene burden test using GWS from > 700,000 individuals. There are several novel interesting findings for BMI and T2D and consequences on other metabolic parameters using human association results to validate phenotypes observed in mouse. The results are interesting to a wider audience, when the WGS-based rare-variant-burden results are provided for T2D and BMI genome-wide (data availability).

Data and methodology: The data is UK Biobank and well-known for high quality, as is All-of-Us. The methods are state-of-the-art, and the evidence provided in Figures and Supplementary Tables is convincing and comprehensive.

Statistics and treatment of uncertainties: The appropriate statistics are generated and provided; some overview rather laundry lists of statistics in results text (first part of results, see details below) might be helpful to the reader. There is no critical appraisal of potential uncertainties in the definition of T2D or CKD and how this might affect results in UKB, where individuals are rather healthy.

Response: Uncertainty in case identification is expected to weaken the strength of any true effect rather than causing inflation or bias. Furthermore, confirmation in the completely independent All of Us sample provides strong confidence that the findings are robust. We have added a sentence to the Discussion on this issue.

“Importantly, we demonstrate that our findings from UK Biobank are robust and reproducible, as several were replicated in an independent US population-based

study (All of Us) which has considerably different demographics, notably its younger age, higher baseline prevalence of T2D and enhanced ethnic diversity³⁶. Replication of this nature is particularly important given the well-recognised healthy volunteer bias that exists in UK biobank.” (Page 13, Line 322-327)

Conclusions: A replication of the association of IRS-2 with CKD independent of T2D and all downstream mediator analyses in All-of-Us, which has probably more and clinically better-defined T2D and CKD, could support the findings and the strong statements made in the discussion. UKB individuals are rather healthy and CKD and T2D are probably (not clear from methods) both made on blood biomarkers rather than a clinical diagnosis. This is OK for association analyses, but the conclusions refer to clinically diagnosed patients, which might be a bit of a stretch as it stands now.

Response: Our case definitions of CKD and T2D in both UKBB and All of Us are based on clinical diagnoses in routine healthcare data, rather than simply biomarker-based definitions. This has been clarified in the Methods text and Supplementary Tables as described above.

We have also now leveraged a recently available PheWAS conducted in the All of Us study (now presented in **supplementary table 13**) to provide orthogonal validation of our findings in an independent cohort:

“Finally, we sought to demonstrate the robustness of this observation with orthogonal validation in an independent cohort. Therefore, we undertook a lookup of IRS2 in a publicly accessible PheWAS of the All of Us Cohort (see methods) finding nominally significant, highly ranked associations for a biomarker of renal function (blood urea nitrogen), chronic kidney disease and other traits related to renal failure (Supplementary Table 13).” (Page 9, Line 222-227)

Clarity and context: The title does not appear to catch the key approach and result of the paper. The manuscript is about gene-burden tests based on WGS addressing coding regions that helps prioritize genes for metabolic health. Is this cardio-metabolic or just metabolic? There is nothing about CAD or CVD. Is “novel insights” and the rest of the title precise enough to distinguish this paper from others and to catch the key approach and message of the manuscript?

Action: As suggested, we have changed the title to **“Population-scale gene-based analysis of whole genome sequencing provides novel insights into metabolic health”**

Specific suggestions for improvements: Abstract and introduction are fine.

Results:

(1) Please provide participant characteristics in a Table for the analyzed UKBB individuals and All-of-Us including stats on the distribution of age, sex, ancestry, BMI, %T2D, how T2D was defined in UKBB (state that All-of-Us use medical records

for T2D), distribution of eGFR, %CKD (and how CKD was defined). Is All-of-US based on hospital records (or from general practitioner)?

Response: We have now added these details in **Supplementary tables 16 and 17**. We have provided code lists for UKBB phenotype definitions in **Supplementary tables 10**. We have detailed the specific concept IDs

(2) Lines 100-103: “we observed stronger associations using WGS than ... using WES, ... 29% increase in mean chi-square”: this is methodologically interesting. Please clarify the sample size (#individuals) for this comparison in the text (here it appears to be different sample size). **A clearer presentation of this comparison (not only by lists in Supplementary Table 2) might be scatter plots of effect sizes (beta or OR, respectively), their standard (SE) standard errors, and #variants for each identified gene burden.** A sole comparison of the change in Chi-Square statistics does not help dissect power by smaller SEs versus power due to larger effect sizes or power due to larger number of variants in the burden test.

Response: We made this claim by comparing the current WGS results to our two previously reported analyses of BMI and T2D using UKBB WES data; the sample size for BMI was **419,668** and for T2D was **418,436**. As suggested, we have conducted more detailed analyses and scatterplots to dissect the factors that have led to greater power for WGS vs. WES. We present the suggested plots below (and included as **Supplementary figures 1 & 2**).

The increased discovery with WGS analysis seems due to the larger all ancestry sample (4.6% higher mean chi-square) and the higher number of variants found by WGS (21% higher mean chi-square) and not due to differences in effect estimates.

Supplementary Figure 1 | Increase variants in UKBB WGS analyses drives increased strength of associations compared to UKBB WES data. A-B: Scatterplots showing overall similar effect estimates for significant masks for our current WGS and our previously reported WES analyses for (A) BMI and (B) Type 2 Diabetes. **C:** Scatterplot showing the overall larger number of variants present in WGS compared to our previously reported WES analyses.

(3) This relates to lines 166-183: the authors appear to repeat this comparison now in the same sample size: please state this more clearly adding the sample size in the main text. Line 180: "In contrast, some variants were identified by WES-only": so, on average, WGS included more variants than WES, but WGS also missed some variants? A graphical presentation of these results (details are already in Supplementary Table 2) might be helpful, given that the conclusion ("... quantify the

enhanced coverage of coding variants provided by WGS over WES in UK Biobank”) is a main message.

Response: We have added a plot of *IRS2* to illustrate the greater gene coverage with WGS vs. WES (**Supplementary Figure 2**), similar to that reported more systematically in the “flagship” WGS paper <<https://www.medrxiv.org/content/10.1101/2023.12.06.23299426v1.full-text>>

(4) Lines 92-160: There are several “laundry lists” on identified genes without and with replication, confirmed and known (a lot of stats in parentheses) in the results text (lines 92-160). Detailed results are in Supplementary Table and an overview is in Figure 2: Could beta-estimates for BMI and OR estimates (as numbers) be added to the Figure 2, together with carrier n and case prevalence? And replicated genes marked in bold? Provide a 2nd panel to show previous WES results on these? Then a large part of these laundry lists in the text can be avoided and a much better understanding of the findings (and their novelty) can be appreciated.

Response: As suggested, we have removed the quoted statistics from the text, cite all relevant tables, and highlighted in bold the replicated genes is display items. We tried adding effect sizes and sample numbers to the plots but found this would make these too cluttered.

(5) Lines 120-128: It is very interesting that the authors link rare-variant-burden genes with GWAS loci. Can the authors rule out a shadow effect where the common GWAS-variant tags a haplotype with many of the rare alleles in the burden test? Then the rare-variant-burden can be significant even if the rare variants do not exert a causal effect? Specifically, what is the rare-variant-burden when adjusting for the GWAS lead variant (effect estimate and P-value) compared to unadjusted? Or is it the other way round: does the rare-variant-burden explain the GWAS signal? (i.e. does the GWAS signal disappear when adjusted for the rare-variant-burden).

Response: As suggested, we have added linear mixed models that include polygenic risk scores for T2D and BMI as covariates. Of our 5 confirmed novel gene-disease associations, four were modestly attenuated (but retain exome-wide significance) and one was modestly strengthened (*HNF4A*-T2D). These results are added to Supplementary table 3.

*“We also tested if any of our rare variant discoveries were ‘tagged’ by common variant associations. We generated polygenic risk scores for each trait and included these as co-variates. Of our 5 confirmed novel gene-disease associations, four were modestly attenuated (but retain exome-wide significance) and one was modestly strengthened (*HNF4A*-T2D). These results indicate that the identified rare variant effects on Type 2 Diabetes and BMI are independent of common variants. (Supplementary Table 3)” (Page 6, Line 138-144)*

(6) Supplementary Table 5, cited from line 128: For the region around each rare-variant-burden identified gene, could the variant with the smallest P-value and its P-value from GWAS be added?

Response: As suggested, in this table (now **Supplementary table 2**), we show the identity and P-value of the lead variant in 'test-statistics' section.

(7) Would it be helpful to separate out some of the first chapter (no subtitle) into the primary analyses (identifying novel gene burden) and the more methodological follow-up (LOO sensitivity analyses and the exploratory analyses of comparing WGS versus WES in sample size)?

Response: As suggested, we have restructured the results section and have included a '**Sensitivity analyses**' section.

A phenotypic association scan of identified genes ... novel role for IRS2

(8) Is the IRS2 association stated as beta = -10.42 for eGFR and OR = 4.0 for CKD adjusted for T2D? Could you please provide both the T2D-unadjusted and T2D-adjusted effects estimates and P-values in the main text and Figure 4? (One caveat here is that this reviewer could not see Supplementary Figure 3; nor any Supplementary Figure).

Response: Those estimates were unadjusted for T2D. As suggested, we now also show T2D adjusted analyses (**Supplementary Table 12**)

(9) It is an interesting finding to link IRS2 with kidney function independent of T2D. "this association does not simply reflect the consequences of T2D-mediated chronic hyperglycaemia": What exactly is the definition of T2D used here for UKB?

Response: T2D in UKBB was defined using the linkage to hospital episode statistics and death certificates. Participants with recorded ICD10, Read and OPSC4 codes consistent with a diagnosis of type 2 diabetes were designated cases, those without any qualifying conditions were considered controls. The specific criteria are now listed in **Supplementary Table 10**.

(10) To show these results for 3 different equations for eGFR is to be appreciated, but: How was CKD was defined? (eGFR_{crea}<60?) Please add n of CKD cases and controls (to Figure 4 legend). UK Biobank is rather healthy with a small proportion of CKD among individuals. Can this be replicated in All-of-U with probably more and clinically better-defined CKD patients?

Response: As described above, CKD in UKBB was defined using a combination of medical records and death certificates with specific qualifying ICD10, OPSC4 and Read codes (see **Supplementary table 10**) and represents clinically diagnosed CKD (not by categorising a single biochemical measurement). We have added cases/control numbers to **Figure 4** legend.

We have also now leveraged a recently available PheWAS conducted in the All of Us study (now presented in **supplementary table 13**) to provide orthogonal validation of our findings in an independent cohort:

*“Finally, we sought to demonstrate the robustness of this observation with orthogonal validation in an independent cohort. Therefore, we undertook a lookup of *IRS2* in a publicly accessible PheWAS of the All of Us Cohort (see **methods**) finding nominally significant, highly ranked associations for a biomarker of renal function (blood urea nitrogen), chronic kidney disease and other traits related to renal failure (**Supplementary Table 13**).” (Page 9, Line 222-227)*

Evidence of ... *IRS1/IRS2* mediated signalling

(11) Again, it is very interesting to try validated mouse phenotypes by human association data.

a. Figure 5 legends state “*IRS-1* reduces body size with modest effect on blood glucose”, but “height” and “fat free mass” and “T2D” are shown: there are various measures of “body size” beyond height (just call it “height”?), is fat free mass a measure of “body size” or a mixture of muscle and bone? Where is the “modest effect on blood glucose”? does this refer to the association with T2D? but T2D status is not necessarily reflecting blood glucose, if treated. Better stick to the concrete variable name, at least in results text and Figure legends.

Response: The legend text quoted by the reviewer refers to previously reported findings in mouse studies with which we compare our displayed human genetic results. We have clarified this distinction in **Figure 5** legend.

b. Also “*IRS-2* causes severe hyperglycaemia” without affecting body size: the term “cause” is difficult in epidemiological studies (please avoid); the association is with T2D, which typically includes antidiabetically treated (and not hyperglycemic at the study center assessment).N Is there still no association of *IRS1* on T2D when adjusted for height and fat-free-mass?

Response: Again “loss of *IRS-2* causes severe hyperglycaemia” refers to previously reported findings in mouse studies. We have clarified this in **Figure 5** legend.

Regarding model adjustments, the addition of lean mass (but not height) as a covariate does strengthen the association of *IRS1* with T2D, but it remains substantially weaker than for *IRS2* and T2D (see below), so this distinction between *IRS1* and *IRS2* remains valid (consistent with findings in mice).

Gene	Outcome	Conditioned_on	Log(OR)	SE	P-Value	n_carriers
IRS2	T2D	NA	1.87	0.29	6.69E-11	58
IRS2	T2D	Fat Free Mass	1.87	0.30	6.97E-10	58
IRS2	T2D	NA	1.87	0.29	6.69E-11	58
IRS1	T2D	NA	0.57	0.22	0.01	172
IRS1	T2D	Fat Free Mass	1.09	0.23	1.78E-06	172
IRS1	T2D	NA	0.57	0.22	0.01	172

UBR2 and UBR3

(12) The OR for T2D for UBR3 was 3.4 in UKB without adjustment for BMI ($P=6 \times 10^{-9}$) and 2.5 ($P=2.7 \times 10^{-4}$) with adjustment: could you re-state the unadjusted results? In All-of-Us, the OR was 2.7 without adjustment: what is it in All-of-Us with adjustment for BMI? Is the definition of T2D in All-of-Us different from the definition in UKB?

Response: As described in the Methods, the definitions used differ slightly as All of Us phenotypes are curated using OMOP. The OR for *UBR3* after adjustment for BMI in All of Us is 1.97.

(13) There are many interesting results but a condensed view in a Figure on these results on UBR2 and UBR3 would be helpful (similar to the Figure for IRS1 and IRS2).

Response: As suggested, we now provide an additional Figure highlighting interesting findings for *UBR2* and *UBR3* on childhood adiposity (**Figure 6**). We also provide updated text citing this Figure and contrasting the apparent effects of these homologs:

“Alongside an increased risk of Type 2 Diabetes, the risk of a clinical diagnosis of hypertension was significantly increased in carriers of UBR2 and UBR3 PTVs in our PheWAS analysis, but the effect observed in UBR3 PTV carriers was nearly double that of UBR2. Intriguingly, UBR3 appears to increase adiposity from an early age, an effect not apparent for UBR2 in UKBB. It is interesting to speculate that differential regulatory roles of these proteins throughout the life course may underlie the heterogeneity in their effects on cardiometabolic risk (Figure 6). ” (Page 12, Line 309-315)

Methods:

(14) There is a generic explanation of the All-of-Us T2D definition in the methods (page 22, line 546: “concept ID 201826...”), but this is the specification of a code, but not what this means in terms of: HbA1c? anti-diabetic medication? Hospital or GP-based medical record of T2D diagnosis? What does “initial entry” mean (line 548): is this not BMI from height and weight measured at study center baseline and the concordant T2D status at the same time? If BMI and T2D records are not at the same timepoint, how was this accounted for in the analyses?

Response: In All of Us, clinically diagnosed Type 2 Diabetes cases were ascertained from medical records using the relevant diagnostic codes within OMOP. For ease of reference, we provide a direct link to the All of Us data browser in the methods where the full list of codes can be reviewed. BMI was also extracted from medical records as All of Us did not measure this in a baseline assessment. We have clarified the methods:

“BMI data were derived from the “body mass index (BMI) [Ratio]” metric (Concept Id 3038553) within the “Labs and Measurements” domain. BMI values <10 or >100 were excluded and the earliest remaining value recorded and corresponding age was used. The “Type 2 diabetes mellitus” identifier (Concept Id 201826, <https://databrowser.researchallofus.org/ehr/conditions/201826>) in the “Conditions” domain facilitated the identification of T2D cases, the age corresponding to the earliest diagnosis of Type 2 Diabetes was used. The participants’ ages were calculated by subtracting the birth year from the timestamp of the earliest record. Among these individuals, 32,462 were identified as T2D cases, and 186,553 served as controls. Only subjects aged over 18 were included in the analyses. Only a small proportion of episodes that indicated a diagnosis of type 2 diabetes had a contemporaneous BMI measurement. As such, to adjust Type 2 Diabetes for BMI we used two approaches – the median BMI value recorded was included as a co-variate in the model or the BMI record closest to Type 2 Diagnosis was used.” (Page 26, Line 614-628)

(15) What IS the definition of T2D in UKB: HbA1c? anti-diabetic medication? how was CKD defined?

Response: In UKBB, both CKD and Type 2 Diabetes were clinically diagnosed cases, ascertained from electronic health records (Hospital episode statistics, death registry, primary care records) using the relevant diagnostic codes (ICD10 or Read code) listed in **Supplementary table 10**. Neither medication or biochemistry were used to define cases.

Discussion:

(1) page 12, line 328: what do you mean with “algorithmically defined CKD”?

Response: Definition of CKD based on linked healthcare data is now more clearly explained in ‘**Phenotype preparation in UKBB**’ with the qualifying codes presented in **Supplementary table 10**.

(2) Link of IRS-2 to SIRD: was IRS-2 associated with parameters of insulin resistance? (lines 345-352). This reviewer might have missed this; maybe this can be made a bit clearer in the results or discussed why no association with insulin resistance was found.

Response: Given the key role of IRS2 in insulin action, we think disposition to insulin resistance is likely. However, it is challenging to test this in UKBB due to the lack of

fasting samples. Proximal defects in the insulin receptor cascade (e.g. dominant negative INSR mutations) cause marked insulin resistance with acanthosis, skin tags, PCOS (in females) and high fasting and post-prandial insulin, but not the dyslipidaemia associated with 'typical' insulin resistance (so TG and HDL are not informative).

To address this question, we have now examined data from ALSPAC, a birth cohort with WES and fasting insulin measurements. We found two probands with non-extreme fasting insulin levels (<97.5th centile and lower than an accepted clinical cut off for severe insulin resistance). Thus, IRS2 haploinsufficiency does not appear to cause early onset severe insulin resistance, however a predisposition to more moderate insulin resistance cannot be excluded. While the diabetes risk in IRS2 variant carriers is likely due to both insulin resistance and beta-cell dysfunction, on reflection we agree the evidence for SIRD are speculative and we have removed it and comment on this issue in the discussion.

Reviewers' Comments:

Reviewer #1 (Remarks to the Author):

The authors met an extensive and comprehensive set of questions and comments across the reviews. They have thoughtfully and thoroughly addressed these issues to better and more accurately address each of their topics of interest. I have little doubt the work will be of interest both mechanistically/biologically and in more directly addressing the comparative value of WES/WGS analyses. I commend the team for their detailed approach to managing and addressing the extensive reviews.

I have no outstanding issues or concerns with this work.

We thank the reviewer for their comments and for highlighting the relevance of our work across a breadth of topics.

Reviewer #2 (Remarks to the Author):

Co-reviewed with Reviewer #1.

Reviewer #3 (Remarks to the Author):

The authors have done a great job addressing the reviewers' questions. I think the more in-depth analysis of the reasons behind the higher yield of signals from WGS vs WES is an important addition. The 4.6% increase from increased sample size and 21% from more variants seems consistent with other reports that WGS results in about 10% increase in information, given that most of the increase in sample size comes from more diverse samples...so the two are related. The change in tone in the title also makes sense given the main message is about new genes not so much WGS vs WES or other approaches.

We appreciate the reviewer's recognition of our efforts and interest in our work.

Reviewer #4 (Remarks to the Author):

Zhao et al. "Population-scale gene-based analysis of whole exome sequencing provides novel insights into metabolic health"

The authors have answered to all comments. However, two concerns remain.

One concern is the still not fully clear description of the outcome variables:

Regarding the definition of CKD in UKBB, the authors answer to a first comment, that hospital episode statistics were used. However, CKD does not typically lead to hospitalization. Hospital records are insufficient for this. To a later comment and methods text, it seems that they used UKBB records from general practitioners (GPs), which makes sense. However, the methods' text on how their main outcome variables were generated does not allow for an understanding of what these outcome definitions means and how well ascertained these can be deemed. The same applies to the diagnosis of T2D. For T2D, this seems to be similar for All-of-Ups but was not fully clear. The CKD finding could have been replicated in All-of-Ups, as the data is available in that study, but this was not done? The term "algorithmically derived CKD" used at very instances in the text is uninformative, as indicated previously by this reviewer, but is still used.

We apologise if this was unclear. The methods section refers to a supplementary table which details the >100 codes we used as qualifying terms to define a diagnosis of Type 2 Diabetes and Chronic Kidney disease. We hope this provides sufficient information for readers to scrutinise and replicate our phenotype definition. However, in response to the reviewers comment we have added a narrative summary of our phenotype derivation to help facilitate partial appraisal of our phenotype definitions while reading the paper without referring to code lists. The term 'algorithmically derived' has been removed.

"Binary outcomes were prepared using a combination of hospital episode statistics (UKBB showcase IDs: 41202, 41204, 41200, 41210) primary care records (UKBB showcase IDs: 42040), death certificates (UKBB showcase IDs: 40001, 40002) and self-reported medical conditions (UKBB showcase ID: 20002). Qualifying codes pertaining to each condition are listed in Supplementary table 10. Any participant with a qualifying code was considered a case, those without a qualifying code considered as controls. For Type 2 Diabetes, qualifying terms included codes specifying diagnoses of non-insulin-dependent diabetes, Type 2 Diabetes and insulin treated type 2 diabetes. Participants who self-reported a history of Type 2 Diabetes were also classified as cases. For chronic kidney disease, diagnostic codes included those specifying chronic renal failure, chronic renal impairment, chronic kidney disease, end stage renal failure, hypertensive renal disease with renal failure or codes indicative of preparation or receipt of renal replacement therapy. Participants who self-reported renal/kidney failure, dialysis or procedures to prepare for peritoneal or haemodialysis were specified as cases. For type 2 diabetes and chronic kidney disease phenotype definitions, all participants who did not meet the qualifying terms were classified as controls, please refer to supplementary table 10 for a full list of qualifying codes."

To the question about the potential impact of misclassification in T2D or CKD, the authors answer that "uncertainty in case identification is expected to weaken the strength of any true effect rather than causing inflation or bias": this is true when the misclassification is in the outcome, but this is not true when the misclassification is in

the covariable used to rule-out a mediating effect. A critical appraisal of the limitations in their case/control definitions and how this may affect their results and conclusions is still lacking.

We thank the reviewer for this comment. The fidelity of our T2D definition is demonstrated by the association of our exome-wide significant masks with glycaemic traits which are measured in the vast majority of the UKBB cohort and unlikely to be subject to ascertainment bias. Although not stated in our manuscript, the validity of our CKD definition is supported by significant associations with predicted damaging mutations in known monogenic causes of CKD including PKD1, PKD2, UMOD and HNF1B. The agreement between findings based on our CKD definition and multiple biochemical surrogates of renal function further supports the validity of our definition. We acknowledge that the above factors do not speak to the NPV of our definition and agree with the reviewer that misclassification of Type 2 Diabetes cases as controls may limit our ability to exclude chronic hyperglycaemia as a mediator of the association between CKD and PTVs in IRS2. However, we do not think it is likely that Type 2 Diabetes mediates the effects of PTVs in IRS2 on eGFR/CKD. Principally, and as presented in Figure 4b in the manuscript, we generally do not observe strong associations with other masks associated with Type 2 Diabetes and renal function, including masks with much stronger effects on Type 2 Diabetes risk. This includes predicted damaging mutations in genes known to cause monogenic diabetes syndromes where diabetic nephropathy is a recognised complication if blood glucose control is suboptimal (e.g. HNF1A). Moreover, we see a similar pattern of association when we use HbA1c measured at UKBB study visits as an index of chronic hyperglycaemia (see below). In addition, when we use HbA1c measured in UKBB as an alternative index of chronic hyperglycaemia and include it as a co-variate in our model we still observe significant associations with CKD ($P=0.005$).

These results strongly support our assertion that the effect of IRS2 PTVs on CKD exceeds that which we would expect if this association was mediated entirely by Type 2 Diabetes/chronic hyperglycaemia. Nevertheless, in response to the reviewers comments we have highlighted how misclassification of T2D cases as controls would affect our results, so readers are aware of this potential limitation.

“While our findings support the notion that IRS2 PTVs cause CKD independent of chronic hyperglycaemia, it is important to note that any misclassification of Type 2 Diabetes cases as controls would bias us towards this conclusion. However, it seems unlikely that Type 2 Diabetes/chronic hyperglycaemia mediates the effects of IRS2 PTVs on CKD as we have not found similar associations for several genes with stronger effects on Type 2 Diabetes risk, including HNF1A where microvascular complications are known to develop if glycaemic control is suboptimal.”

The second concern refers to the problem that rare-variant-burden can arise from a shadow effect by a common variant associated with the trait that resides in or near the gene in question. When rare alleles sit on a haplotype of a common variant allele, and that common variant is associated with the outcome, the rare-variant-burden can be significant, but this does not support the gene as causal. It is mandatory to exclude a shadow effect and only genes where the rare-variant-burden remains significant after adjustment can be deemed to support the gene as causal. The authors deemed their analysis to adjust for common variant effects as “sensitivity analyses”, but it is more than that: the unadjusted analyses can be just a first indicator; the adjusted analyses are the valid analysis.

While we appreciate the reviewer’s perspective, we prefer to evaluate the role of common variation downstream and note there is no set community standard for this approach. In our experience it is rarely the case that very rare variant signals from gene burden testing are driven by common variants. We use a number of downstream diagnostic approaches that evaluate this, including leave-one-out models and synonymous burden masks. Furthermore, the reviewer will note in our response to the query below that we found no evidence of UKBB-specific WGS common variant signals impacting any of our signals.

Even more of concern is the authors approach of how to adjust the rare-variant-burden for a shadow effect. They use a polygenic risk score (made across some common variants associated with the outcome). However, some polygenic score for the outcome is not sufficient to use here: the region (with their WGS-data) needs to be tested for single-variant-association and the significantly associated variants from this need to be used for the adjustment to rule out a shadow effect of the rare-variant-burden. The polygenic score might or might not include any variant of that region; and very likely does not contain all the variants that are associated with the outcome in single-variant testing. Thus, the rare-variant burden of the 21 claimed genes and particularly the three genes newly identified for BMI and the seven genes newly

identified for T2D needs to be substantiated by adjusting for variants associated with the respective outcome in single-variant testing. As the results stand, it is not clear whether their claimed genes are truly supported as causal genes by rare-variant-burden or mere shadow effects.

We have now performed the analysis described by the reviewer and do not find any meaningful attenuation of effect after adjustment for any regional common variant associations identified directly from UKBB whole genome sequencing data. Briefly, for each novel, replicated gene associated with T2D or BMI we examined WGS data for regional ($\pm 500\text{kb}$ from index gene), common variants ($\text{MAF} > 0.001$) significantly associated with the test trait ($P < 6.15 \times 10^{-7}$). In total 6 variants met these criteria (3 independent signals near HNF4A, 1 near RMC1, 1 near TNK2 and 1 near IP6K1). To demonstrate our rare variant gene burden associations were independent of these signals we included these variants in a generalised linear model with the cognate rare variant gene burden mask and calculated attenuation in test statistics compared to the same model without common variants. These results show no meaningful changes in test statistic post conditional analysis, supporting our original interpretation. We have now added these results to the text and supplementary table 20-21.

“We performed additional analyses adjusting for regional, common single variant associations identified directly in UKBB WGS data ($\text{MAF} > 0.001$, $P < 6.15 \times 10^{-7}$) and did not observe any meaningful attenuation of test statistics (Supplementary Table 20-21).”

Phenotype	Gene	Mask	Beta	SE	P-Value	Beta_conditioned	SE_Conditioned	P-value conditioned
T2D	HNF4A	MISS_REVEL_05	0.43	0.06	2.70E-12	0.46	0.06	6.11E-14
T2D	HNF4A	MISS_REVEL_07	0.42	0.08	3.49E-07	0.48	0.08	6.70E-09
T2D	IP6K1	MISS_REVEL_05	1.34	0.26	1.39E-07	1.35	0.26	1.34E-07
T2D	RMC1	HC_PTV	1.01	0.22	2.85E-06	1.01	0.22	2.87E-06
BMI	TNK2	HC_PTV	0.81	0.18	6.64E-06	0.81	0.18	6.56E-06

We thank the reviewer for their thorough critique and for the opportunity to improve our manuscript.